# Effect of the Dry-Wet Cycle on the Performance of Marine Waste Silt Solidified by Calcium Carbide Residue and Plant Ash

Hao Yang, Jianfeng Zhu *, Yanli Tao, Zhengqing Wang and Qiqi Zheng

School of Civil Engineering and Architecture, Zhejiang University of Science and Technology, Hangzhou 310023, China
* Correspondence: zhujianfeng@zust.edu.cn

**Abstract:** This research aims to investigate the potential of engineering waste marine silt stabilized by a self-developed stabilizing chemical additive called PZ-1 as a subgrade filler. PZ-1 is composed of calcium carbide residue (CCR) and plant ash (PA) under an optimal composition ratio determined by coupling particle swarm optimization with a support vector machine (PSO-SVM). The effect of curing agent dosage ($w_g$), temperature ($w_T$), number of dry-wet cycles ($N_{dw}$), and organic matter content ($w_o$) on the micro-macro behavior of the stabilized silt were investigated via the unconfined compressive strength (UCS) test, the scanning electron microscope (SEM) test, and the X-ray diffraction (XRD) test. The experimental results demonstrate a significant positive effect of PZ-1 on the unconfined compressive strength ($q_u$) of marine engineering waste silt with curing agent contents of 0~8%. It was also found that strength improvement of the stabilized silt can be attributed to the formation of gelling substances such as C-S-H and calcite. The water resistance of the stabilized silt can be enhanced by increasing the dosage of the curing agent. Moreover, the organic matter content and ambient temperature have significant effects on the dry-wet cycle tolerance of solidified soil, among which temperature exhibits a more obvious impact.

**Keywords:** calcium carbide residue; plant ash; dry-wet cycle; waste silt

## 1. Introduction

A huge amount of marine waste silt is generated during engineering constructions, such as deep foundation pit engineering and subway tunneling construction in the coastal cities of China [1–3]. These silts have the typical characteristics of high water content, high compressibility, and low shear strength, which determine the fact that reasonable treatments are essential before any engineering construction. The traditional disposal method of discharging the waste silt into the ocean had been banned by the Chinese government because of the detrimental and polluting effects on the marine ecosystem. Meanwhile, the reuse of waste silt has received wide attention in the context of resource scarcity. Therefore, an urgent need exists to develop an efficient, economical, and reasonable methodology to dispose of and further reuse the engineering waste silt.

The stabilization technology of mixing solid or liquid binders with silt has been highly valued and is claimed to be very effective for the treatment of engineering waste silt. The solidified silt can be used as roadbed filler to realize the reuse of engineering marine waste silt. At present, the most popular stabilized agents in soil improvement are lime, Portland cement, and their compound types [4–7]. By adding stabilized agent, the strength of the soil is enhanced due to the volcanic ash reactions and the hydration between calcium hydroxide, alumina, silica, and water [8–12]. Unfortunately, the production of cement, lime, and their composite types is an energy-intensive process. Specifically, to produce 1 ton of cement, 0.95 tons of carbon dioxide was discharged into the air (about 5% of the global anthropogenic $CO_2$ emission) and 5000 MJ of energy was consumed [13]. Similarly, the production of 1 ton of lime corresponds to 0.79 tons of carbon dioxide discharge and 3200 MJ

of energy consumption [14]. Furthermore, cement production accelerates the depletion of natural resources, with the generation of 1 ton of cement consuming about 1.5 tons of limestone and clay [15,16]. As a result, low-carbon and green strategies are crucial for the sustainable development of the stabilized agent industry [17]. In this respect, using various types of solid wastes as industrial waste and agricultural waste to substitute for high carbon-emission stabilized agents, such as lime, Portland cement, or their compound types, was considered a fairly effective strategy [18,19].

Calcium carbide residue, CCR, is the main byproduct of acetylene production, which mainly contains calcium hydroxide, $Ca(OH)_2$. It is estimated that the annual production of calcium carbide slag in China exceeds 20 million tons [20]. The stacking of CCR will occupy a large number of valuable land resources and further pollute the surrounding soil, water, and organisms due to its high alkalinity [21]. Meanwhile, CCR has been identified as an ideal substitute for quicklime due to its high content of CaO and thus is considered a promising soil adhesive [22–28]. Plant ash (PA) is the combustion residue of plants such as herbs and woody plants and is mainly the byproduct of biomass power generation technology. With a high purity of amorphous $SiO_2$ with significant volcanic ash activity, PA had been considered one of the most promising renewable resources. Thus, PA is believed to be of great potential for application in the construction industry, in line with the concept of sustainable green building [29–31].

At present, some researchers have used CCR and PA as raw materials for marine soft soil curing treatment, and the physical and mechanical behaviors of stabilized soil have been addressed in detail [32–35]. However, the long-term performance of the stabilized soil, such as durability under dry-wet cycles, and the effect of soil pH during the test have not been studied, which greatly restricts the application of solidified waste silt as subgrade soil In fact, in practical engineering, the subgrade soil faces the risk of damage or destruction in a dry-wet environment. Dry-wet cycles can alter the physical and mechanical properties of the soil by changing its microstructure and pore spatial distribution. The marine silt contains organic matter. The presence of organic matter in the silt during the solidification of the sludge will inhibit the solidification reaction, but the degree of inhibition is not known. Therefore, it is urgent to investigate the physical and mechanical responses of the curing soil under dry-wet cycles and the effect of the organic matter content on the solidification effect of solidified soil under dry-wet cycles, and to judge the feasibility of the developed binder of CCR and PA in marine waste silt solidification.

In the present research, a self-developed curing agent PZ-1, with CCR and PA as the main raw materials, was adopted to solidify the marine engineering waste silt. The optimal ratio of the two constituents was determined by the PSO-SVM algorithm. Unconfined compressive strength tests (UCS) and dry-wet cycle tests were performed to investigate the mechanical properties of stabilized soil. X-ray diffraction (XRD) and scanning electron microscopy (SEM) were used to comprehensively study the microstructural properties of the stabilized soil. The influence of the organic matter content and the temperature on the curing effect of PZ-1 under different dry-wet cycles were explored in detail. Additionally, the curing mechanism of CCR and PA was obtained by investigating the chemical and physical reactions, such as the pozzolanic reaction, the carbonation reaction, and the ion exchange reaction inside the solidified soil.

## 2. Materials and Methods

### 2.1. Materials

The raw materials used in the experiment are mainly marine engineering waste silt, CCR, and PA, as shown in Figure 1a–c, respectively. The marine waste silt was excavated from an engineering foundation pit with 6 m in buried depth in Ningbo, China. The fundamental physical and engineering properties of the marine silt are listed in Table 1.

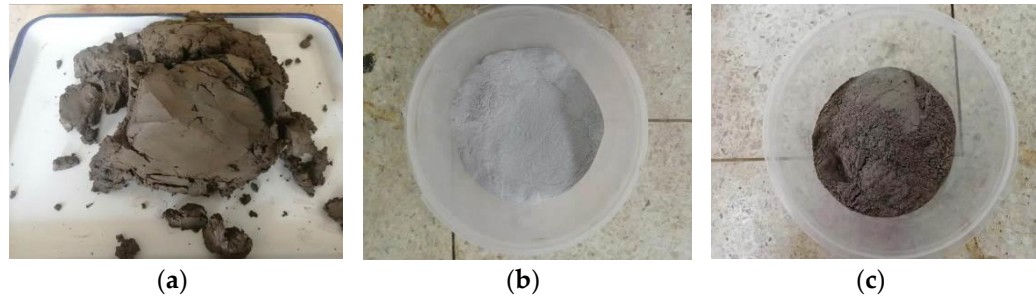

|  (a)  |  (b)  |  (c)  |

**Figure 1.** Main raw material of (**a**) waste silt, (**b**) CCR, and (**c**) PA.

**Table 1.** Physical properties of the waste silt.

| Property Classification | Value |
| --- | --- |
| Natural water content (%) | 45.5 |
| Initial void ratio | 1.32 |
| Wet unit weight (kN/m$^3$) | 17.2 |
| Liquid limit (%) | 36.2 |
| Plastic limit (%) | 22.6 |
| Plasticity index (%) | 13.6 |

The grain size distribution curves of the CCR and PA obtained via sieve analysis and that of the marine silt obtained using a hydrometer are presented in Figure 2. As shown in Figure 2, the marine waste silt has the smallest average particle size among the three materials with a maximum particle size of less than 0.1 mm. The CCR occupies the largest average particle size with the maximum particle size of far more than 2 mm.

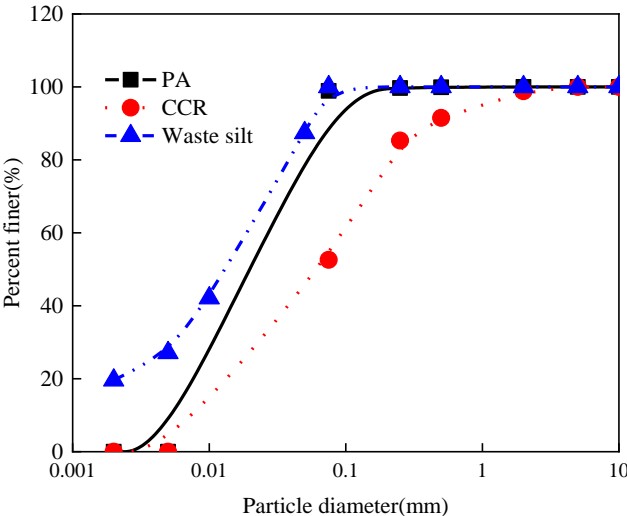

**Figure 2.** The grain size of silt PA, CCR, and waste silt.

### 2.2. Test Plan

As shown in Tables 2 and 3, the UCS and SEM tests of PZ-1 solidified silt with different curing agent contents, $w_g$, were carried out to address the effect of $w_g$ on the unconfined compressive strength of the stabilized soil, $q_u$. Moreover, the chemical compositions of silt, CCR, PA, and PZ-1 solidified silt were detected by XRD (shown in Table 4). As shown in Table 5, the behavior of the solidified soil with different numbers of dry-wet cycles, $N_{dw}$; PZ-1 contents, $w_g$; organic matter contents, $w_o$; and different reaction temperatures, $w_T$, was investigated in detail.



**Table 2.** Experimental design of UCS.

| Group Numbers | $w$/% | $T$/d | $w_g$/% |
|---|---|---|---|
| U1 | | | 0 |
| U2 | | | 2 |
| U3 | | | 4 |
| U4 | 16 | 28 | 6 |
| U5 | | | 8 |
| U6 | | | 10 |

**Table 3.** Experimental design of SEM.

| Group Numbers | $w$/% | $T$/d | $w_g$/% |
|---|---|---|---|
| S1 | | | 0 |
| S2 | | | 2 |
| S3 | | | 4 |
| S4 | 16 | 28 | 6 |
| S5 | | | 8 |
| S6 | | | 10 |

**Table 4.** Experimental design of XRD.

| Symbol | Test Material |
|---|---|
| X1 | Marine soft clay |
| X2 | PA |
| X3 | CCR |
| X4 | Stabilized silt by PZ-1 |

**Table 5.** Experimental design of dry-wet cycles.

| Group Numbers | $w$/% | $T$/d | $w_g$/% | $w_T$/°C | $w_o$/% | $N_{dw}$ |
|---|---|---|---|---|---|---|
| DW1 | | | 0 | | | |
| DW2 | | | 2 | | | |
| DW3 | | | 4 | | | |
| DW4 | 16 | 28 | 6 | 60 | 2, 4, 6, 8, 10 | 0, 1, 3, 6 |
| DW5 | | | 8 | | | |
| DW6 | | | 10 | | | |
| DW7 | | | 0 | | | |
| DW8 | | | 2 | | | |
| DW9 | | | 4 | | | |
| DW10 | 16 | 28 | 6 | 30 | – | 0, 1, 3, 6 |
| DW11 | | | 8 | | | |
| DW12 | | | 10 | | | |

*2.3. Methodology*

2.3.1. Organic Matter Test

The organic matter test was carried out using an HH–SA oil bath, as shown in Figure 3a–d. The organic matter content in marine engineering waste silt was determined by the potassium dichromate volumetric method in view of JTG 3430-2020 [36], under the condition of an oil bath of 170~180 °C, the organic carbon in the silt was oxidized by a specific amount of a potassium dichromate-sulfuric acid solution. The use of strong oxidizers for potassium dichromate and sulphuric acid during heating oxidizes the organic carbon. Then, the remaining potassium di-chromate was titrated with a ferrous sulfate standard solution to estimate the organic carbon, and a control group without silt in the test tube is required to determine the consumption of sodium dichromate used to oxidize organic carbon by comparison. The soil organic matter content was calculated by multiplying soil

organic carbon by the "Van Bemmelen factor" of 1.724, which assumes that soil organic matter contains 58% carbon.

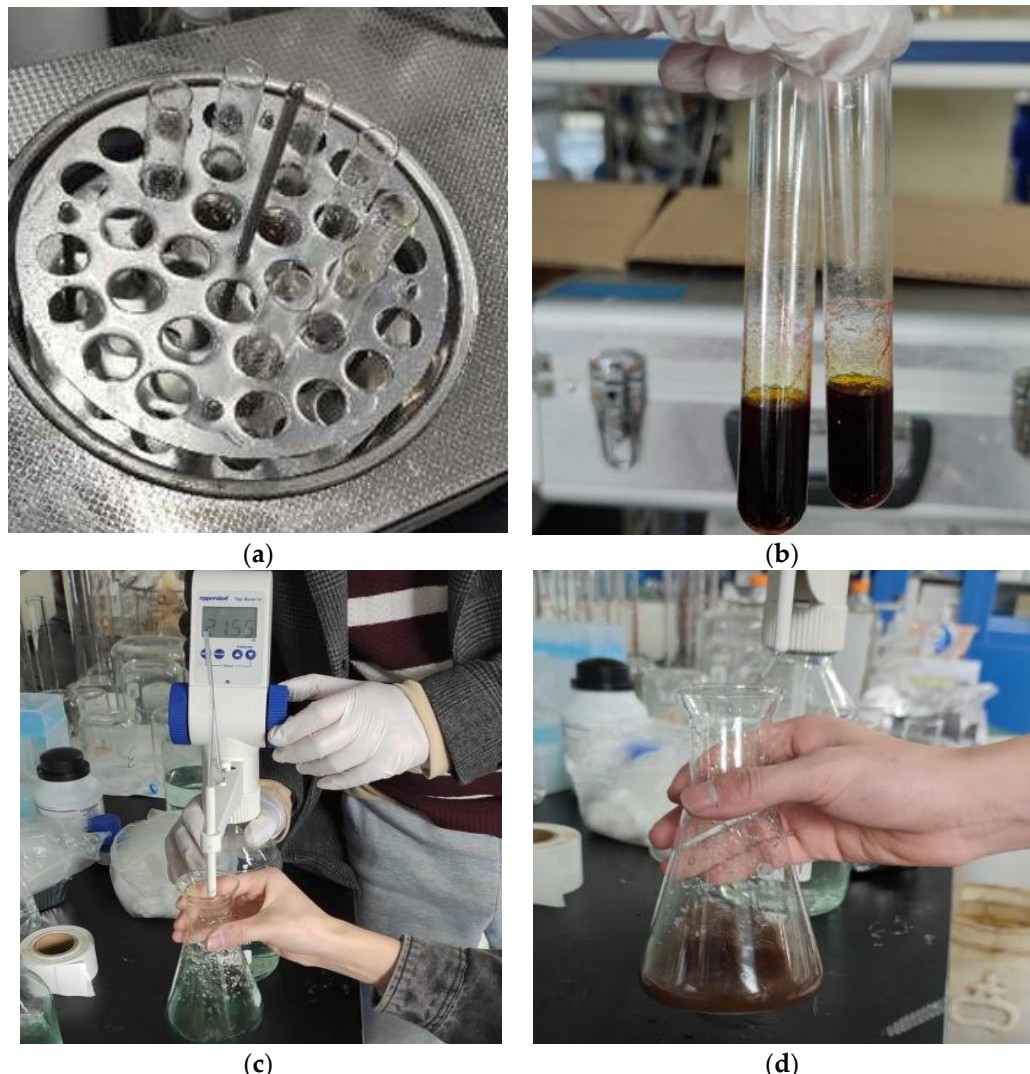

**Figure 3.** The procedure of organic matter test of (**a**) oil bath, (**b**) mixture after oil bath, (**c**) potassium dichromate titration, and (**d**) ending of titration.

### 2.3.2. Compaction Test

As shown in Figure 4a–f, a D075-light compaction instrument was adopted to carry out the compaction test, and the procedure can be listed as follows:

(1) The marine engineering waste silt was dried, broken into pieces, and then passed through a 40 mm sieve.

(2) Soil samples with water contents of 12%, 14%, 16%, 18%, and 20% were prepared by even and good mixing for backup, respectively.

(3) The net weight of the compaction cylinder with an inner diameter of 102 mm and a height of 116 mm was weighed and numbered as $m_1$, and then, the compaction cylinder was placed on hard, horizontal ground.

(4) A thin layer of Vaseline was wiped around the inside wall of the compaction cylinder, and a plastic film was placed on the bottom plate.

(5)  A specific amount of dry soil with the designed water content was poured into the compaction cylinder in three layers, during which the soil surface was flattened and compacted layer by layer. During the compaction process, it should be noted that the hammer is free to plumb down, and the hammering point should be evenly distributed on the soil surface. The height of each layer after compaction should be roughly equal, and the soil surface at the interface of the two layers should be roughened to increase the interlock of each layer.

(6)  After compaction, the sample height beyond the top of the compaction cylinder should be removed. After cutting and digging along the inner wall of the casing with a soil cutter, the casing was twisted and taken down to measure the super-elevation and average the multiple measured values to ensure that the value was less than the specification limit of 6 mm.

(7)  Carefully trim the sample along the tap top and remove the bottom plate. When the bottom surface of the sample exceeds the outside of the tube, it should be repaired.

(8)  Clean the outer wall of the cylinder, weigh it, and set the mass to $m_2$. Then, two soil samples of 15~30 g were taken from the center of the soil mass, and the moisture content of the soil ($w$) was measured in parallel. As the cylinder volume, $V$, is constant, the density can be calculated by the following equation:

$$\rho = \frac{(m_2 - m_1)}{V} \tag{1}$$

The dry density of the soil is given by:

$$\rho_d = \frac{\rho}{1 + w} \tag{2}$$

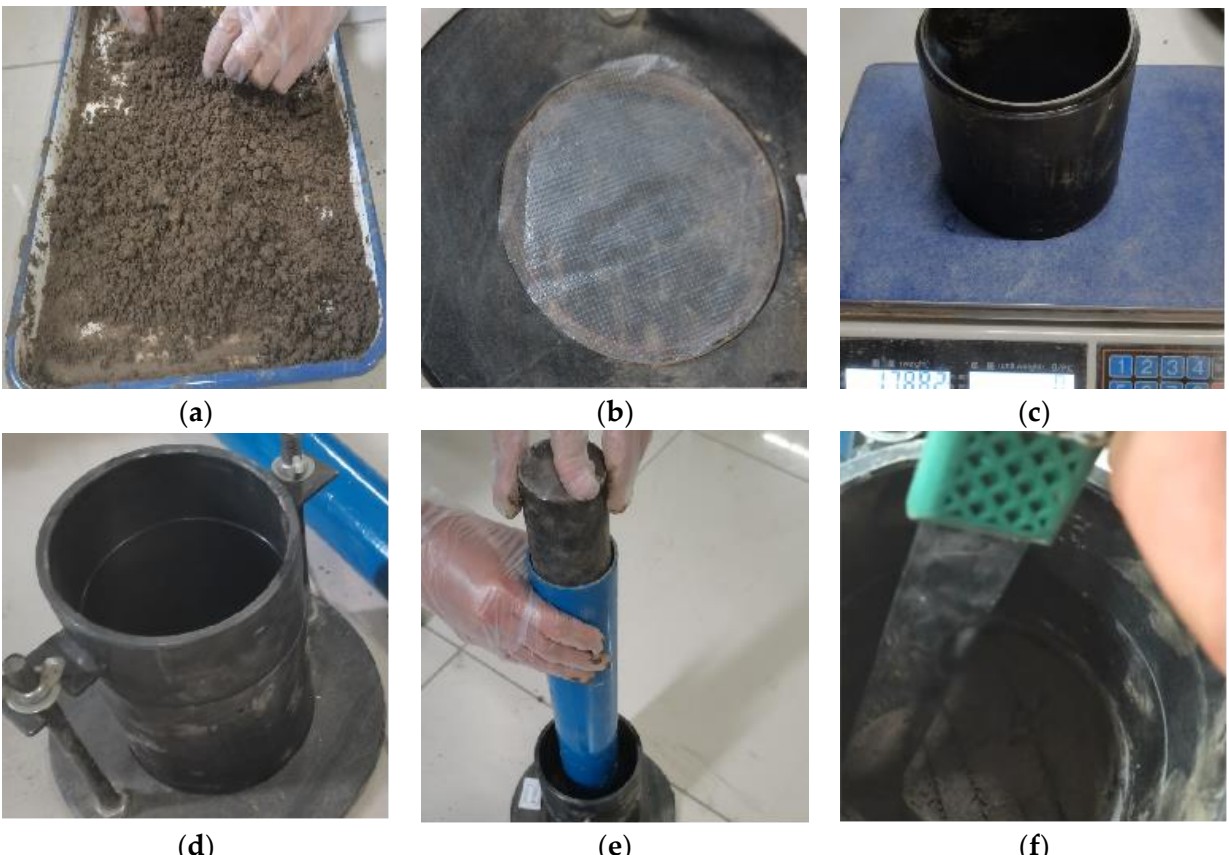

(**a**)          (**b**)          (**c**)

(**d**)          (**e**)          (**f**)

**Figure 4.** *Cont.*

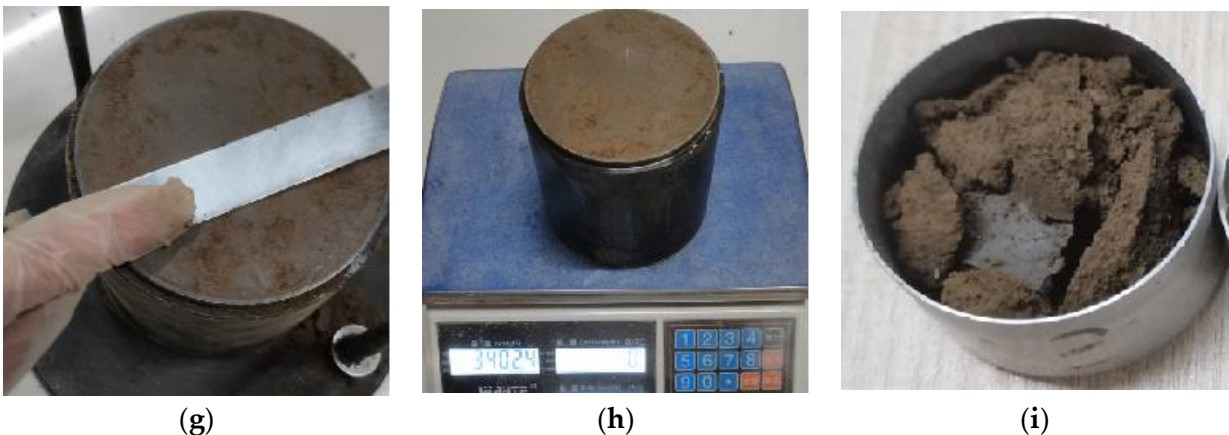

(**g**)             (**h**)             (**i**)

**Figure 4.** Compaction test of (**a**) soil sample preparation, (**b**) net weight weighing of compaction cylinder, (**c**) placing plastic film on the spacer, (**d**) assembly of instrument, (**e**) compaction, (**f**) layered shaving, (**g**) topsoil sample troweling, (**h**) weighing, (**i**) water content measuring.

### 2.3.3. Solidified Soil Specimen Preparation

In the preparation process of the stabilized soil samples, the marine engineering waste silt was oven-dried, smashed, and sieved with a 2.00 mm sieve. Then, the pre-weighed PA and CCR were added to the marine engineering waste silt in dry conditions; the required devices are shown in Figure 5a–e.

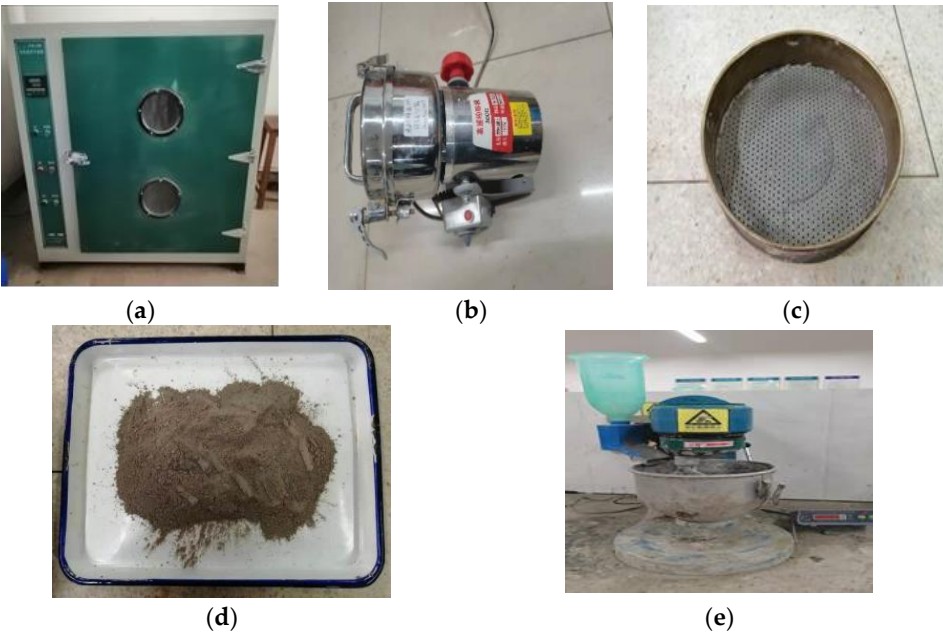

(**a**)           (**b**)           (**c**)

(**d**)           (**e**)

**Figure 5.** Remolded soil preparation instruments of (**a**) oven, (**b**) grinder, (**c**) sieve, (**d**) crushed dry soil, and (**e**) cement mortar mixer.

Figure 6a–e display the static pressure specimen preparation process. As shown in Figure 6, the homogeneous mixture was first put into a cylindrical mold with a diameter of 39.1 mm and a height of 80 mm four times by using the quasi-static compaction method (QSCM). At each time, the loading frame was used to statically press the soil sample into a specified height, and the roughening treatment measure was adopted to interlock the interface of each layer until the specimen preparation was completed and pushed out. The prepared samples were then placed in the SHBY-40 standard constant temperature and humidity curing box. The relative humidity was maintained at 95 ± 3%, and the

temperature was maintained at 20 ± 2 °C. After 28 days of curing, the stabilized silt samples were taken out for testing, with the specimens with obvious changes in volume density or obvious surface cracks removed.

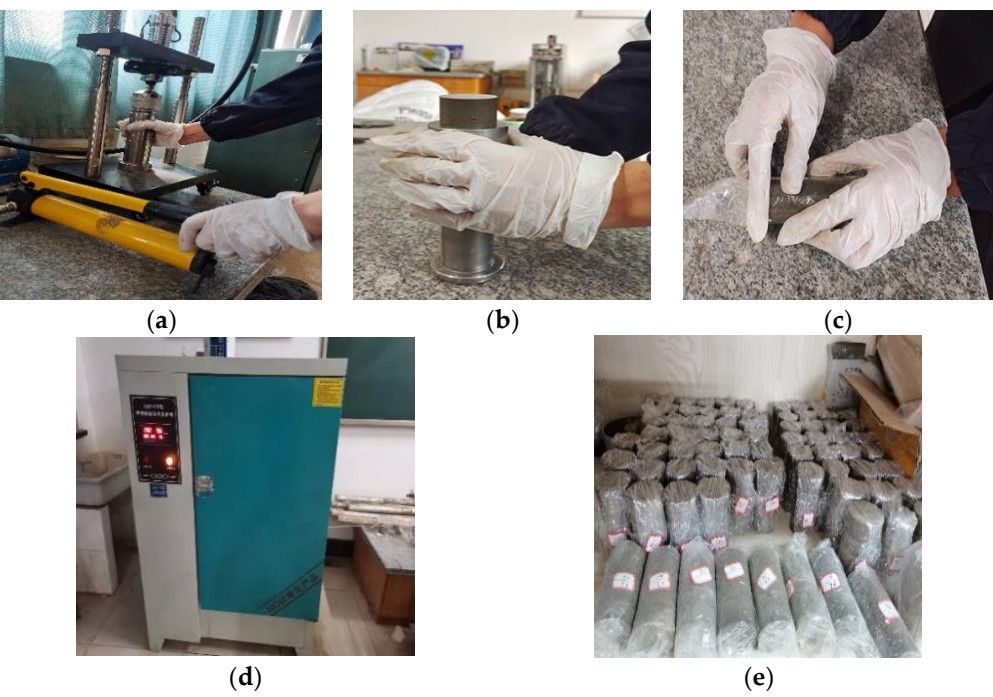

**Figure 6.** Static pressure specimen preparation steps of (**a**) compressing soil, (**b**) specimen taking out, (**c**) specimen packaging, (**d**) specimen curing, (**e**) formed specimen.

### 2.3.4. Dry-Wet Cycles

Figure 7a–f present the dry-wet cycle operation process. As shown in Figure 7, the soil specimens, after being cured for 28 days in the standard curing box, were placed in an oven with a temperature of 60 ± 3 °C and left static for 24 h. Subsequently, the specimens were placed in a constant temperature environment for 1 h, and then, water was added to the outside beaker until the specimens were submerged. After that, the beakers containing the specimen were put in the standard curing room (temperature 20 ± 3 °C, humidity 95%) for 23 h. Finally, the specimens were taken with tweezers and were washed to eliminate surface debris with distilled water. In this way, a single time of the dry-wet cycle test was completed. Then, the specimen was taken out and put into another beaker to continue the next dry-wet cycle until all the specified cycles of the 6 required in ASTMd4843-1998 were finished.

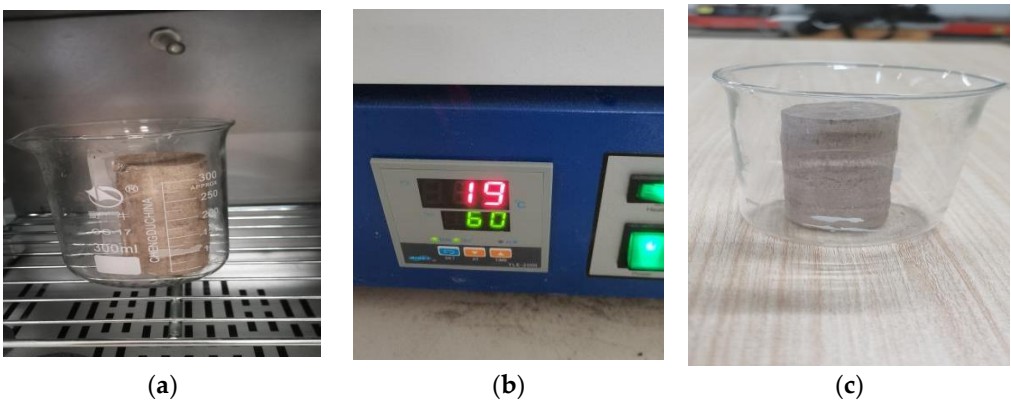

**Figure 7.** *Cont.*

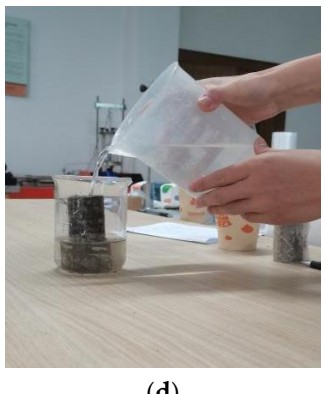 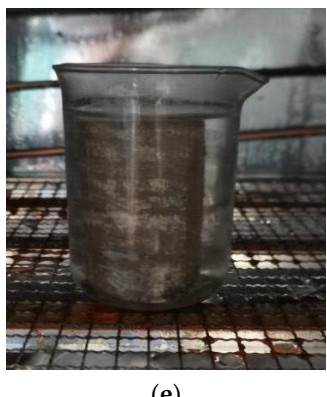 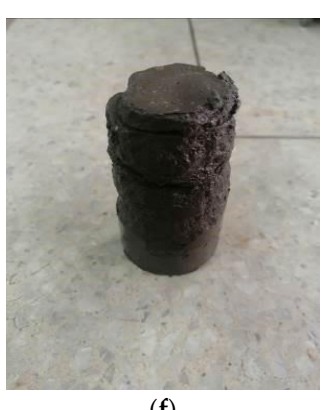

|  |  |  |
|:---:|:---:|:---:|
| (**d**) | (**e**) | (**f**) |

**Figure 7.** Dry-wet cycle operation of (**a**) 24 h in circumstances of $60 \pm 3\,°C$, (**b**) temperature setting, (**c**) constant temperature curing, (**d**) adding water, (**e**) constant temperature and humidity curing, and (**f**) surface observing.

### 2.3.5. Scanning Electron Microscopy Test

The SU-70 electron microscope scanner produced by Hitachi was used to address the surface morphology and composition analysis test system. The microscopic morphology, fracture, electronic structure, crystal structure and internal structure, electric field, or magnetic field of the material surface were observed, and then, the composition of the microscopic area of the material surface was quantitatively and qualitatively analyzed.

### 2.3.6. X-ray Diffraction (XRD) Test

XRD is a method of measuring the crystal structure of a substance by crystal X-ray diffraction to analyze the phase. The fully automatic X-ray powder diffractometer (model D8 ADVANCE) produced by the Bruker Company of Germany was used in the experiment. The diffraction patterns of the related materials were analyzed; the composition of the materials was obtained, and the chemical reactions of the solidified soil were analyzed.

### 2.3.7. Unconfined Compressive Strength Test (UCS)

Unconfined compressive strength is the fundamental and representative performance index of solidified soil. In this experiment, unconfined compressive strength was adopted as one of the main evaluation indexes to judge the performance of marine waste silt and the stabilized soil. The test instrument involves the E45 microcomputer-controlled electronic universal testing machine produced by Mets sans Group in the United States.

Before the test, a thin layer of Vaseline was coated on the upper and lower ends of the specimen and placed at the center of the lower-pressure plate. The upper-pressure plate was slowly dropped until touching the upper surface of the sample, followed by the setting of the parameters and the clearing of the readings of the displacement table and the shaft pressure. The displacement control mode was adopted to maintain a loading speed of 1 mm/min for the upper-pressure plate.

The unconfined compressive strength ($q_u$) can be determined by averaging the results of the three parallel soil samples in which the data exceed 15% of the average strength and will be judged to be useless.

### 2.3.8. PSO–SVM

With the aim of acquiring the optimum proportions between CCR and PA for marine waste silt, two groups of UCS tests were conducted. In the first group, termed as the single mixed test, the effects of CCR and PA on soil improvement were investigated, and their preliminary dosage ranges were determined, respectively. In view of this, the second group of tests in the central composite orthogonal design, termed as the double mixed tests, were performed to ascertain the optimum ratio of additives using the PSO-SVM algorithm. As

shown in Figure 8, the SVM model plays the role of predicting the $q_u$ of stabilized soil at a specific curing age in the PSO-SVM system. Additionally, the PSO algorithm was adopted to search for the optimal ratio of the combined stabilized agents. Detailed information about the function and application of the PSO-SVM can be found in Zhu et al. [37].

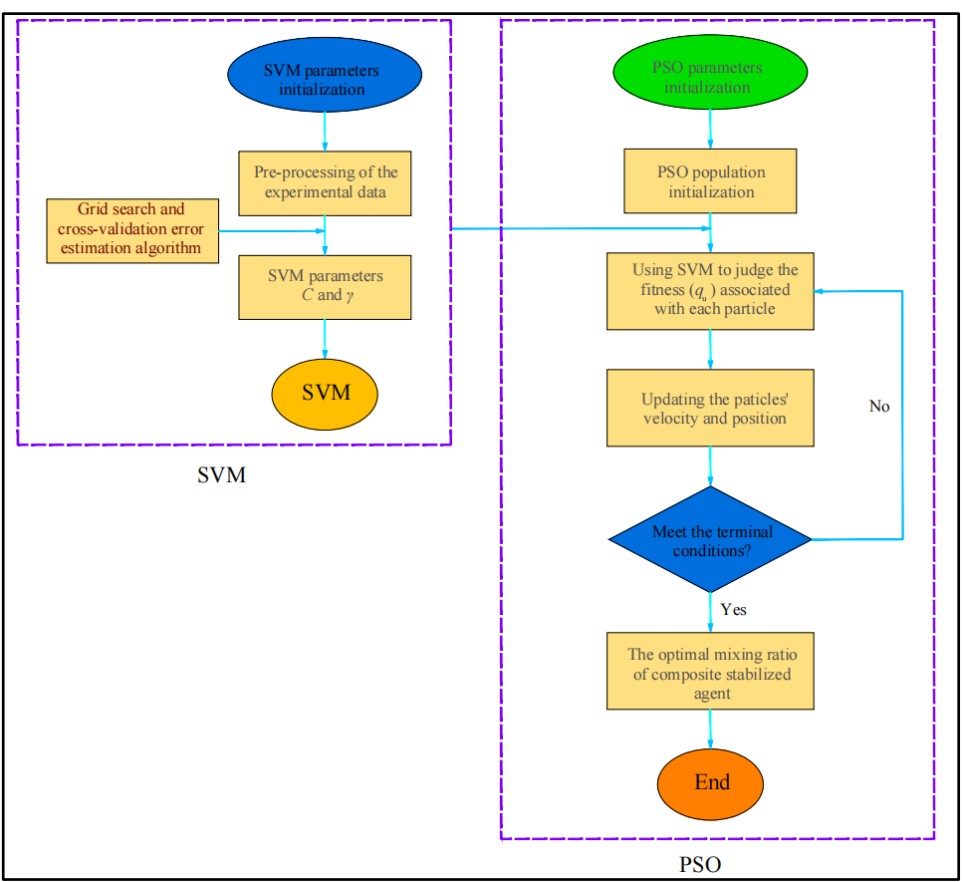

**Figure 8.** Flow chart of PSO–SVM for the optimal mixture ratio of the composite curing agent [37] (Reproduced with permission from Jianfeng Zhu, [Construction and building materials]; published by [ELSEVIER SCI LTD], [2022]).

2.3.9. Image Processing

To investigate the microstructure of solidified marine waste silt quantitatively, micro-parameters, such as the abundance, roundness, and morphological fractal dimension, were obtained using the image processing technology with the 6.0 version of Image-Pro Plus (IPP). The micro-mechanism of strengthening marine waste silt with the developed composite curing agent was further addressed.

(1)    Abundance (*C*)

As defined in Equation (3), *C* refers to the ratio of the short axis to the long axis of the soil particles in the measurement window and can reflect the shape characteristics of the soil particles in the two-dimensional plane. As a result, *C* ranges from 0 to 1. When *C* tends to 0, it indicates that the particles tend to be long strips. In contrast, if *C* approaches to 1, it indicates that the particles have become round.

$$C = \frac{B}{L} \tag{3}$$

where *B* and *L* present the minor and major axis lengths of the soil particles, respectively.

(2)　Roundness (*R*)

As an important indicator of the particle shape, *R* represents the shape factor in a two-dimensional shape measurement method. It can be adopted to judge the roundness degree of the soil particles. Similarly to *C*, the value of *R* also ranges from 0 to 1. A larger value of *R* demonstrates a rounder shape of the soil particle [38–42]. *R* can be defined as

$$R = \frac{4\pi A}{P2} \tag{4}$$

where *A* and *P* are, respectively, the area and perimeter of the soil particles.

(3)　Fractal dimension (*D*)

Because of the heterogeneous property of soil, it is difficult to quantitatively describe the geometric characteristics of the soil particles and soil pores. In view of this, the fractal theory is developed to analyze the micro-behavior of soil [43]. In the fractal theory, the fractal factor, *D*, presents the main invariant, the complexity, and the irregularity of the fractal objects. At present, *D* is mainly determined by the counting box dimension method [44,45] and the perimeter area method [45–47]. In this study, the perimeter area method was adopted, and the associated formula can be

$$\lg P = \frac{D}{2}\lg A + X \tag{5}$$

where *A* and *D* are, separately, the area and fractal dimension of the soil particles, and *X* is the fitting constant.

The perimeter and area data were extracted in 0, and the double logarithmic plot was drawn to obtain the slope *K* from IPP 6.0; then, the final fractal dimension is given by

$$D = 2 \times K \tag{6}$$

As shown in Figure 9, the slope *K* is obtained as 0.367 by fitting the double logarithm diagram, and the associated value of *D* can be 0.734.

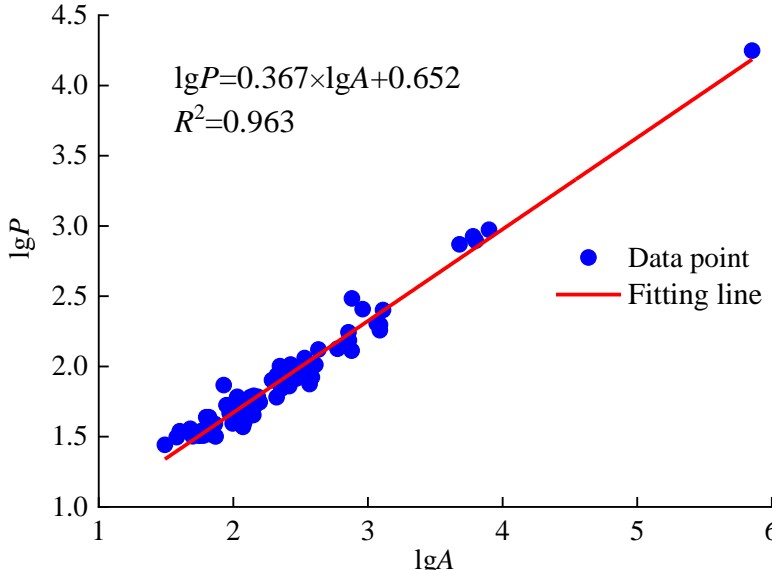

**Figure 9.** The trend line lg*P*–lg*A*.

## 3. Results and Discussion

### 3.1. Optimum Ratio of Additives

The optimal ratio of the CCR and PA addressed by the PSO–SVM is CCR:PA = 13.35:16.06, and the corresponding prediction value (50% of water content, 6% of humic acid content, and 7 days of curing time) of $q_u$ is 226.924 kPa, which is close to the measured $q_u$ (7 days of curing time) of 224.651 kPa, with a relative error of 1.01%. The curing agent for the CCR and PA predicted by the PSO-SVM was named PZ-1.

### 3.2. Organic Matter Content of Marine Waste Silt

As listed in Table 6, the average initial organic matter content, $w_{oia}$, of marine waste silt is merely 0.558%. As a result, its influence on the following preparation of organic silt is believed to be ignorable.

**Table 6.** The organic matter content of waste silt.

| Group Number | $w_{oi}$ | $w_{oia}$ |
|:---:|:---:|:---:|
| 1 | 0.567% | |
| 2 | 0.556% | 0.558% |
| 3 | 0.552% | |

### 3.3. Compaction Test Results

The compaction curve of natural waste silt is shown in Figure 10. As seen in Figure 10, the optimum moisture content of the soil is about 16%, corresponding to a maximum dry density of 1.98 g/cm$^3$. According to Equation (2), the relevant soil density is 2.297 g/cm$^3$, which can be used as the baseline for the following sample preparation.

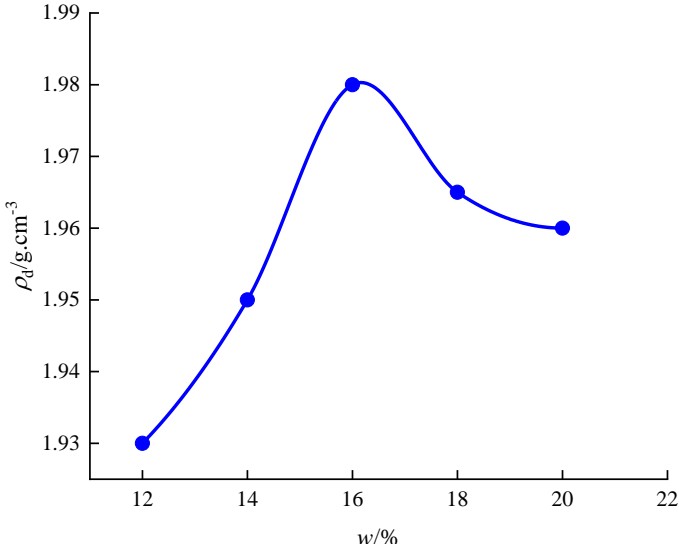

**Figure 10.** Compaction curve of natural waste silt.

### 3.4. Curing Mechanism of PZ-1 Stabilized Waste Silt

3.4.1. Chemical Compositions Analysis

Figure 11a–d clearly show the minerals of the marine waste silt, CCR, PA, and stabilized silt detected by the XRD tests, respectively. It can be seen that the marine waste silt in Ningbo mainly involves minerals such as quartz (SiO$_2$), illite (K(Al$_2$Fe)(Si$_3$Al)O$_{10}$(OH)$_2$), albite (Na(AlSi$_3$)O$_8$), orthoclase(K(AlSi$_3$)O$_8$) and magnesium salts. The CCR is mainly composed of portlandite (Ca(OH)$_2$) and a small amount of calcite(CaCO$_3$) and quartz(SiO$_2$). As can be seen from Figure 11c, the detected X-ray diffraction peaks of PA are mostly calcite (CaCO$_3$), quartz (SiO$_2$), and some potassium salts.

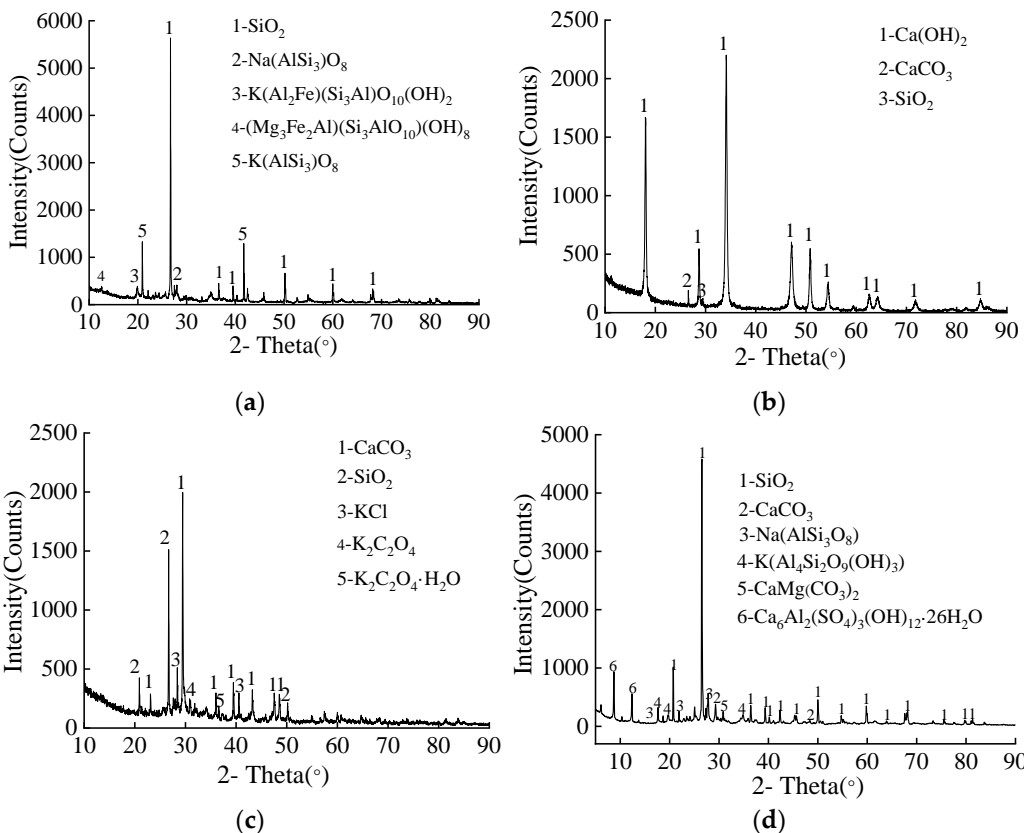

**Figure 11.** XRD analysis of (**a**) soft clay, (**b**) CCR, (**c**) plant ash, and (**d**) stabilized silt.

The XRD patterns of stabilized silt after 28 days of curing are shown in Figure 11d. Figure 11d demonstrates that the main peaks are $SiO_2$, $CaCO_3$, $Na(AlSi_3O_8)$, $K(Al_4Si_2O_9(OH)_3)$, $CaMg(CO_3)_2$, and $(Ca_6Al_2(SO_4)_3(OH)_{12}\cdot26H_2O)$. Accordingly, the stabilized silt mainly contains quartzite, calcite, sodium feldspar, illite, dolomite, and ettringite. Most of $SiO_2$ comes from abandoned soil and PA, and the sodium limestone mainly derives from the waste soil. Calcite mainly comes from the CCR and PA. Additionally, the reaction between $Ca(OH)_2$ and $CO_2$ generated calcite in the stabilized silt. Dolomite may be produced by the multi-reactions between $Mg^{2+}$, $Ca^{2+}$, $CO_2$, and $H_2O$ in the silt. Ettringite can be considered to be one of the productions of the pozzolanic reaction between the CCR and PA. It should be noted that other pozzolanic products, such as C-S-H gel, cannot be identified by XRD in all samples because of its poor crystalline structures; similar results were addressed by Vichan and Rachan [48] as well.

### 3.4.2. Effect of Curing Agent Dosage

As shown in Figure 12, the curing agent dosage, $w_g$, plays a significant role in the strength development of solidified silt. With the increase of $w_g$ from 0 to 8%, $q_u$ increases from 410.354 kPa to 1403.447 kPa with a sharp growth rate of 242.01%, and the relationship between $q_u$ and the content of the curing agent can be expressed by a linear function when the hydration reaction can be fully carried out, which is consistent with the model developed by Prinya et al. [49]. This can be attributed to the hydration of the curing agent which produces C-S-H gel, calcite, and ettringite [48], which further enhances the structure and integrity of the silt, as shown in Figure 13a–c. As a result, the macro-mechanical behavior of silt, such as the unconfined compression strength, is improved. However, when the dosage of PZ-1 is greater than 8%, $q_u$ drops to 1403.447 kPa with a decrease rate of 15% in contrast to the peak value of $q_u$. This could be due to the fact that the water content in the sample is slightly insufficient to promise the hydration of PZ-1, which further reduces the stabilized production of C-S-H, dolomite, and ettringite and weakens

the structure and integrity of the stabilized silt, as shown in Figure 13c. Consequently, the unconfined compression strength of the stabilized silt drops when the agent dosage exceeds a certain value.

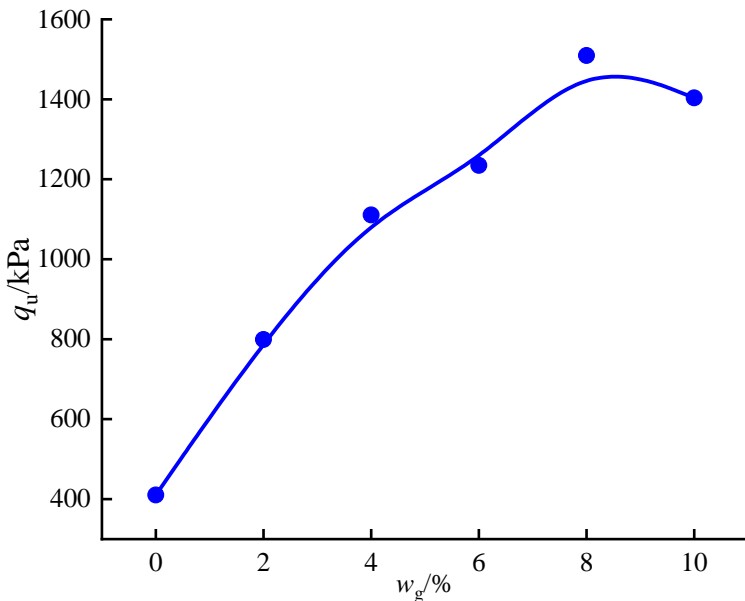

**Figure 12.** Relationship between subgrade filler strength and curing agent PZ-1 content.

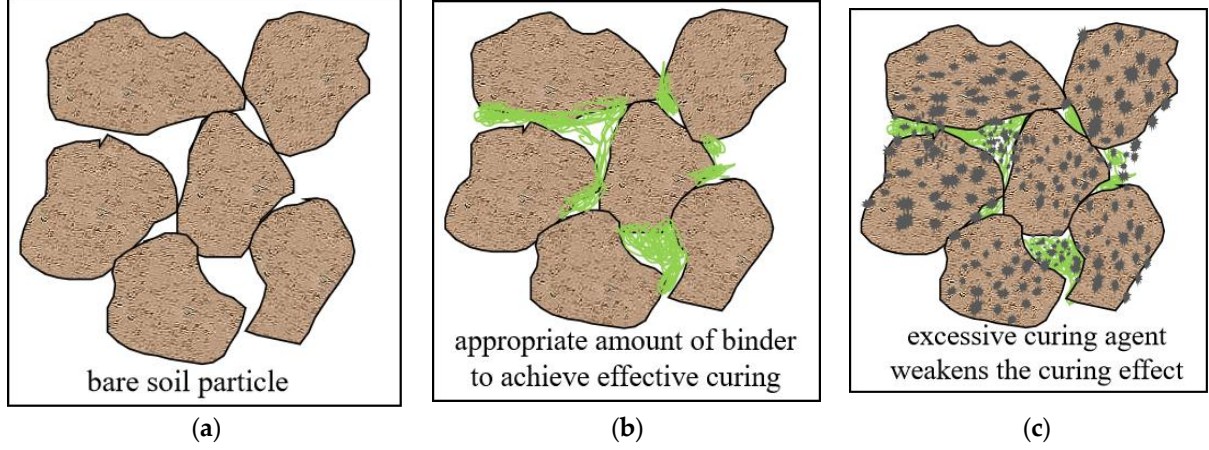

**Figure 13.** Mechanism diagram of remaining curing agent activity in soil: (**a**) before curing, (**b**) appropriate curing agent for curing, (**c**) excessive curing agent for curing.

It should be noted that after adding a mere 2% PZ-1 curing agent, the $q_u$ of the solidified soil with 28 days of curing approaches 800 kPa, which means that the solidified soil is sufficiently applicable in strength for subgrade filler in road engineering.

### 3.4.3. SEM of the Stabilized Silt

To further understand the micro-mechanical variation of the solidified silt with different PZ-1 contents in detail, the scanning electron microscope test ($\times$1 k) of the samples with the dosages of 0%, 2%, 4%, 6%, 8%, 10% were carried out. The results are shown in Figure 14a–f. As seen in Figure 14a–f, the original marine soil displays a loose microstructure and large pores. The contact mode between the soil particles is generally a point-to-surface pattern, as shown in Figure 15a. After adding 2% of PZ-1, as shown in Figure 15b, a relatively dense structure is observed and a bit of gel product, such as C-S-H, is produced, which plays the role of bonding the soil particles together. With the further

increase in the dosage of PZ-1, the solidified soil decreases in the amount of large pores and presents a gradual change of the particle contact mode to a face-to-face pattern (shown in the black circle in Figures 15b and 16). This can be explained by the fact that the hydration of the PZ-1 results in the increase in pozzolanic products such as C-S-H gel and ettringite, which further fill the pores and enhance the density of the soil.

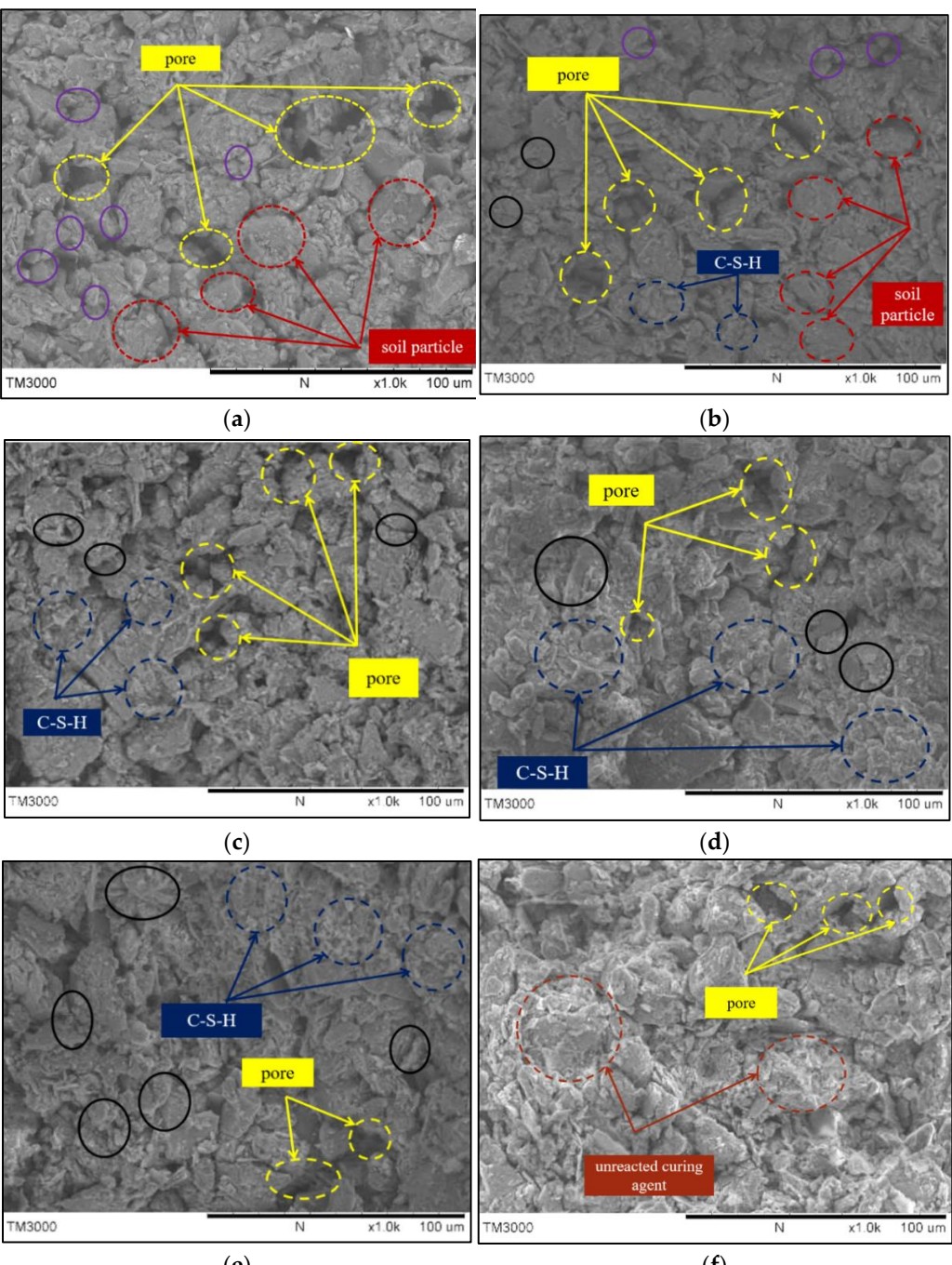

**Figure 14.** SEM micrographs of the stabilized samples: (**a**) $w_g$ = 0%, (**b**) $w_g$ = 2%, (**c**) $w_g$ = 4%, (**d**) $w_g$ = 6%, (**e**) $w_g$ = 8%, (**f**) $w_g$ = 10%.

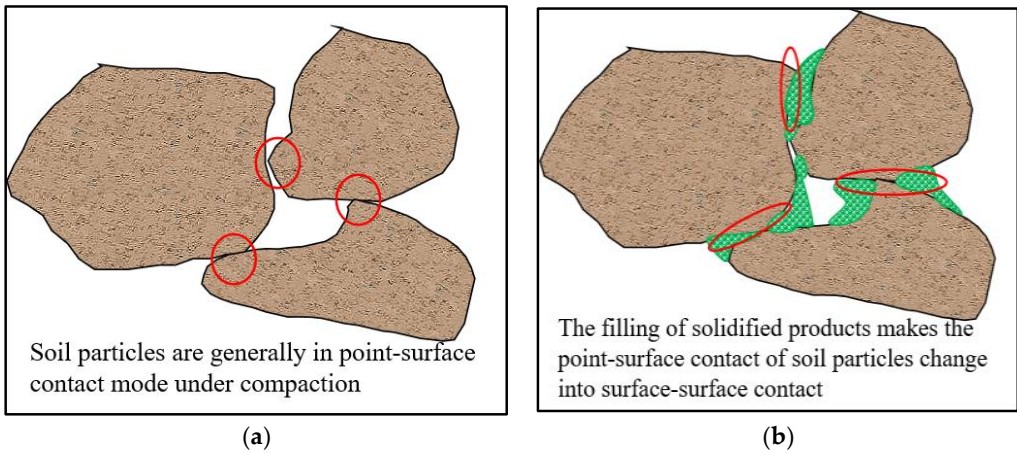

**Figure 15.** Transformation of the contact pattern of soil particles of (**a**) point–surface contact, (**b**) surface–surface contact.

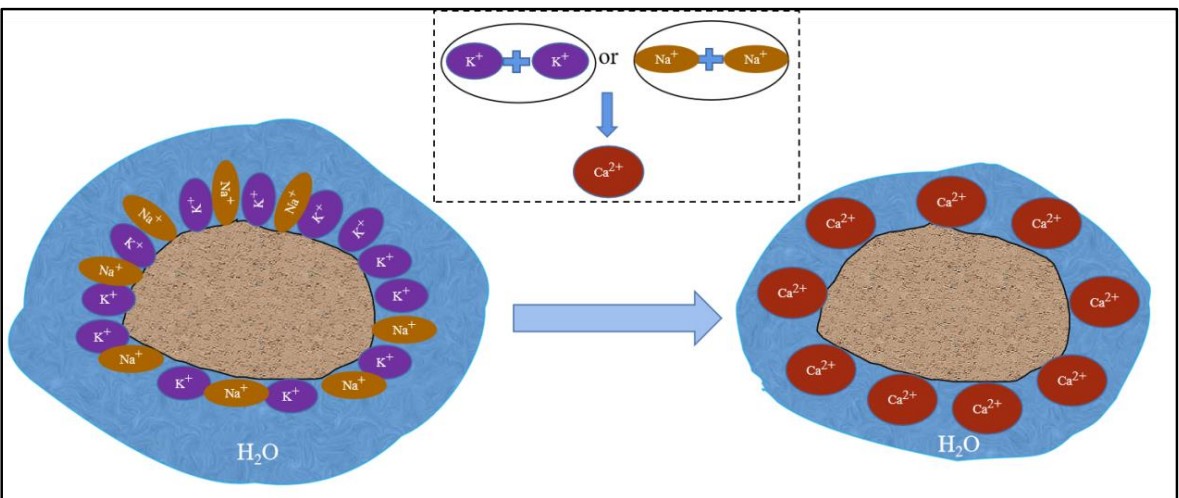

**Figure 16.** Ion exchange reaction.

However, the structure and integrity of the solidified silt are observed to be weakened by the unreacted curing agent (marked with brown circles) on the surface of the soil particles, as shown in Figure 14f, due to the negative effect of a high concentration of PZ-1 on its hydration. This powerfully confirms the fact that the $q_u$ of the stabilized silt with 10% of PZ-1 is lower than that with 8% of PZ-1, as shown in Figure 12.

In addition to the functions of cementing the soil particles and filling the pores of C-S-H gel, the physical effect of the curing agent on waste silt promotes the dense arrangement of the soil particles as well. As shown in Figure 16, the surface of the soil minerals is negatively charged, which will attract low-valence cations, such as potassium ions, $K^+$, and sodium ions, $Na^+$. The calcium ions, $Ca^{2+}$, of the added PZ-1 will exchange equivalently with monovalent cations such as $K^+$ and $Na^+$ in the soil particle surface and further reduce the thickness of the adsorbed water film of soil particles and eventually improve the strength of the marine silt.

### 3.4.4. Microscopic Quantitative Analysis

To further address the micro-behavior of stabilized silt with different dosages of PZ-1 in detail, the IPP 6.0 was utilized to determine the micro-structural parameters of the stabilized silt, such as the abundance, *C*, roundness, *R*, and morphological fractal dimension, *D*, in the following section.

From Figure 17, it can be observed that the distribution of $C$ is mainly concentrated in three intervals, [0.2, 0.4], [0.4, 0.6], and [0.6, 0.8], while there are fewer particles in the range of $C < 0.2$ and $C > 0.8$, with the total proportion of less than 10%. Natural marine silt soil particles tend to be flat, accounting for 40% and 31% in [0.2, 0.4] and [0.4, 0.6], respectively. When the curing agent content is 2%, the shape of the soil particles is mainly elliptical, with the proportions of 38%, 35%, 20% in the ranges of [0.2, 0.4], [0.4, 0.6], and [0.6, 0.8], respectively. With the increase in the dosage of PZ-1, the amount of solidified silt particles in the small abundance range ($C < 0.4$) gradually decreases and that in the large abundance region ($C > 0.4$) gradually increases, which is because the increment in $w_g$ will promote the generation of oblate solidification products such as C-S-H. In this respect, the solidified silt particles tend to be circular as a whole [50,51].

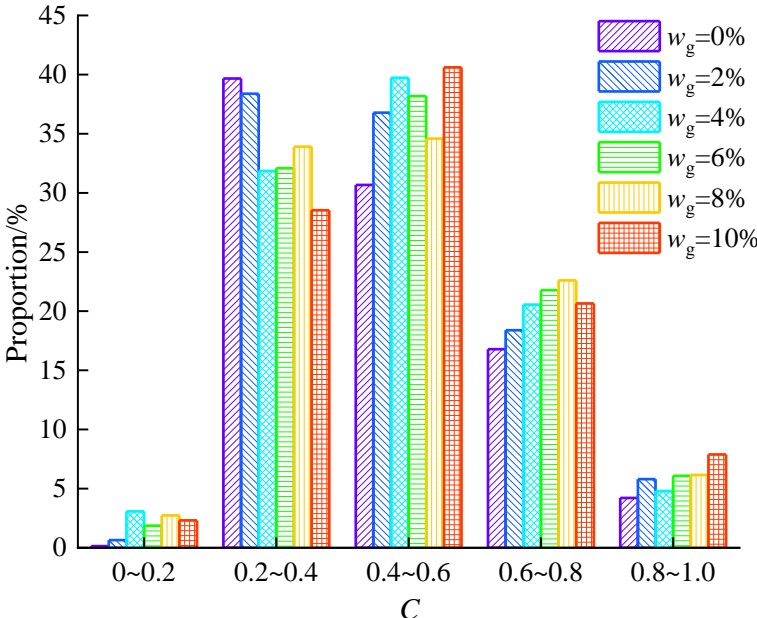

**Figure 17.** Histogram of the micro-particle abundance of solidified waste silt.

As the abundance merely presents the morphology distribution of the stabilized silt particles, the variation law of particle morphology cannot be directly observed. Therefore, it is necessary to use the micro-structural parameter of roundness, $R$, to express the general morphology of the stabilized silt particles under different dosages of curing agents. As shown in Figure 18, $R$ behaves in parabolic rising with the increase of $w_g$, and when $w_g$ is increased to 10%, $R$ is about 0.5011, with a relative increment of 8.44% in comparison to the case when $w_g = 0$%. Consequently, the increase in binder content promotes the development of round particles, resulting in a slight increase in the $R$ of the solidified silt. In addition, circular particles have larger shear stiffness and lower stress state sensitivity than the particles of other shapes, which further improves the mechanical properties of the solidified silt [52,53].

Figure 19 shows the relationship between fractal dimension, $D$, and $w_g$. As seen in Figure 19, $D$ generally reduces with the increase of $w_g$, which is probably due to more hydration products such as C-S-H gel, generated with higher $w_g$ filling the large pores, enhancing the micro-structure of the stabilized silt and improving the orderliness of the silt particles. However, no significant effect of $w_g$ on $D$ is observed considering that a mere 6.28% reduction of $D$ is obtained with a $w_g$ ranging from 0% to 10%.

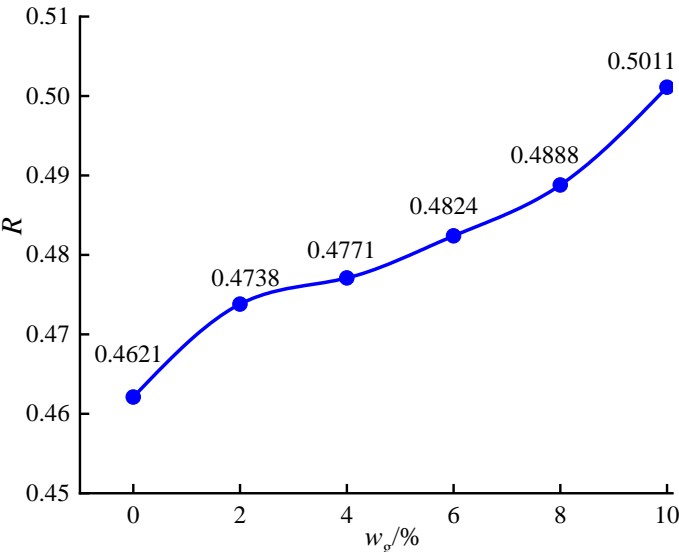

**Figure 18.** Roundness of solidified waste silt microscopic particles.

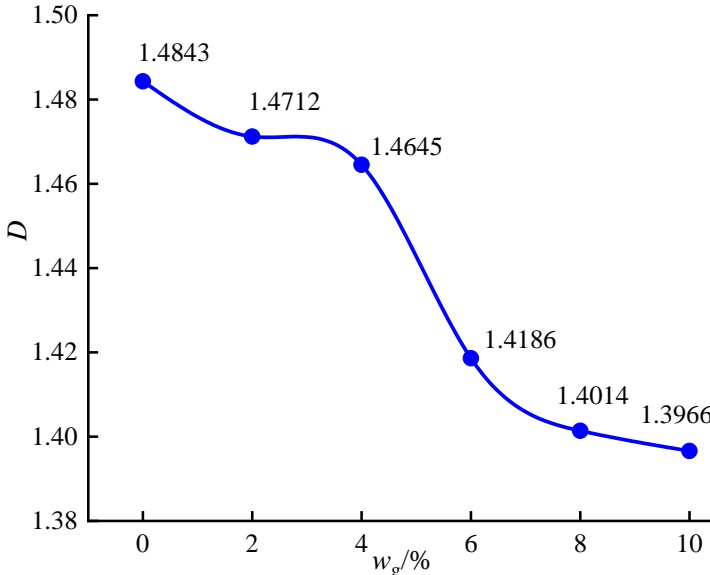

**Figure 19.** Fractal dimension of solidified silt under different dosages of PZ-1.

### 3.4.5. Stabilized Mechanism of PZ-1

The main stabilized mechanism of PZ-1 on waste silt is shown in Figure 20. As seen in Figure 20, the $Ca(OH)_2$ from the CCR can react with the $SiO_2$ from the PA and the active mineral composition from the marine silt to form a cementitious substance such as calcium silicate hydrate (C-S-H), as shown in Equation (7). Moreover, $Ca^{2+}$ and $Mg^{2+}$ will carbonize under an alkaline environment to form $CaCO_3$ and $CaMg(CO_3)_2$, as shown in Equations (8) and (9), to fill pores and cement soil particles and form a whole.

$$Ca^{2+} + 2OH^- + SiO_2 \rightarrow C\text{-}S\text{-}H \tag{7}$$

$$Ca^{2+} + CO_3^{2-} = CaCO_3 \tag{8}$$

$$2Ca^{2+} + Mg^{2+} + 2CO_3^{2-} = CaMg(CO_3)_2 \tag{9}$$

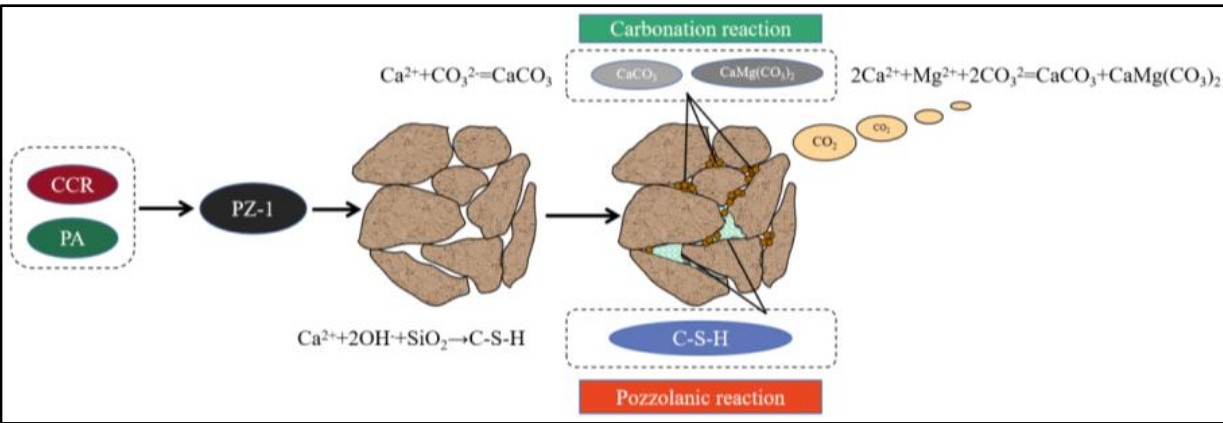

**Figure 20.** Stabilized mechanism of the waste silt with PZ-1 curing agent.

*3.5. Dry-Wet Cycles of Stabilized Silt*

3.5.1. Effect of Dry-Wet Cycles on $q_u$ of Stabilized Silt

As shown in Figure 21, the $q_u$ of the stabilized silt with different curing agent contents, $w_g$, drops gradually with the increase of the dry-wet cycles. Moreover, the $q_u$ of the specimen with a curing time of 28 days conforms to the exponential relationship with the evolution of the dry-wet cycle, which is consistent with the prediction model developed by Ye et al. [18]. When the dosage of PZ-1 is very low, at 2%, $q_u$ behaves in a linear decrease, which could be due to the fact that insufficient solidifying agent incorporation will produce too weak a structure of the solidified silt to suffer the action of one dry-wet cycle.

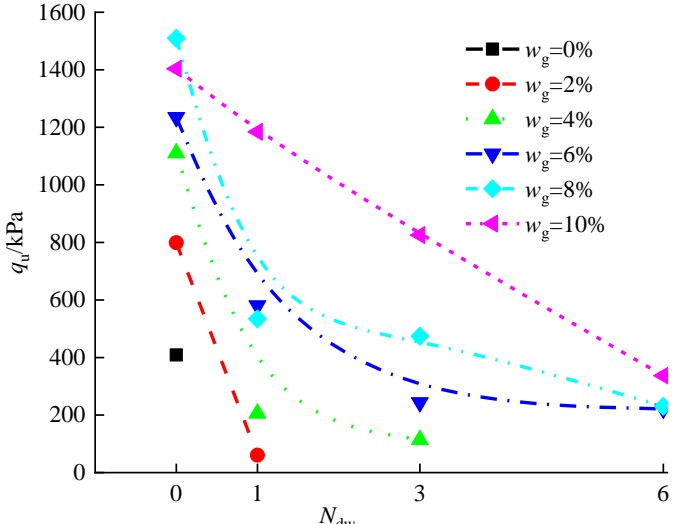

**Figure 21.** Effect of different dosages of PZ-1 curing agent on dry-wet cycle.

When the dosage of PZ-1 is increased to 4%, as presented in Figure 21, the water resistance of the stabilized silt is enhanced by comparison to that with 2% of PZ-1. However, the bonding structure of the stabilized silt is weak as well, and the associated $q_u$ loses 81% in just one dry-wet cycle. As the number of dry-wet cycles increases, $q_u$ drops continuously until by the third dry-wet cycle the specimen becomes too weak to suffer any unconfined compressive loading.

As the adding amount of the composite PZ-1 is increased to 6%, the stabilized silt becomes strong enough to suffer the attack of six dry-wet cycles. In addition, the decreasing trend of $q_u$ demonstrates a flat trend of the solidified waste silt particle after the third dry-wet cycle, indicating that when the solidified agent incorporation is increased to a certain

extent, the mechanical properties of the solidified waste clay tend to be stable after a certain number of dry-wet cycles.

When the dosage of PZ-1 is further increased to 8%, the bonding structure of the solidified silt is further enhanced to suffer the attack of six dry-wet cycles without collapsing. Moreover, it can be observed in Figure 21 that the $q_u$ of the stabilized silt with the $w_g$ of 8% after each cycle is always higher than that of 6%. Consequently, it can be concluded that a higher dosage of curing agent induces a stronger bonding structure of the soil. Additionally, the $q_u$ of the solidified silt decreases by nearly 67% after the first dry-wet cycle test, which is similar to the results of the solidified silt with other amounts of curing agents. As a result, the first dry-wet cycle has the greatest negative impact on destructing the structure of the solidified soil [54,55].

It can be also observed that the original strength of the specimen with a $w_g$ of 10% is lower than that of 8% due to the negative effect of the redundant curing agent, as addressed above. However, with the continuation of dry-wet cycling, the internal unreacted agent has the chance to contact with water, and a series of hydrations are produced in the specimen. Hence, in the first dry-wet cycle test, the $q_u$ of the solidified clay specimen at a $w_g$ of 10% is only reduced by about 20%, with twice the $q_u$ of that of 8%. With the increase in dry-wet cycles, the strength of the solidified silt at the $w_g$ of 10% is still significantly greater than that of less PZ-1 content at the same of $N_{dw}$. As a result, the stabilized silt with PZ-1 behaves with stronger water resistance with the increasing amount of curing agent.

### 3.5.2. Characterization of Morphology of the Stabilized Silt during Dry-Wet Cycles

Figures 22–27 show the morphology of the specimens with different contents of PZ-1 in each dry-wet cycle. On the whole, no significant difference in the shape of the solidified specimen with $w_g$ from 0% to 10% was observed before the dry-wet cycle test. Moreover, the luster of the specimen gradually brightens with the increase of $w_g$. However, with the increase of $N_{dw}$, different degrees of spalling, damage, and cracks were observed in each group of specimens. The damage of the specimen initially emerged in the corner because of the stress concentration and the contact of each layer induced by sample preparation, which indicates that the integrity of the soil in the contact parts of each layer is still weak compared with the other parts of the specimen, though the layered surface of the specimen has been roughened by the roughening tool according to the code of the GBT 50123-2019 Standard for Geotechnical Test Methods. This may be due to the fact that bubbles first flush out from the contact parts and further take out some minerals of the sample and eventually result in microscopic pores and surface damage during the action of the dry-wet cycles.

As shown in Figure 22a,b, the structure of the natural marine silt is destructed completely without suffering any dry-wet cycle, determining that the natural silt has no resistance to the dry-wet cycle. As shown in Figures 23 and 24, after one and three dry-wet cycles the solidified silts with a $w_g$ of 2% and 4% have become eroded with penetrated cracks transversely distributed along the sites of hierarchical compaction. Furthermore, the edge angles of the above two samples have almost disintegrated, which directly results in the reduction in the unconfined compressive strength of the stabilized silt. Figures 25–27 show the morphology of the specimens with a $w_g$ of 6%, 8%, and 10%, respectively. As seen in Figures 25–27, the appearances of the three groups of specimens remain intact without any alteration after the first dry-wet cycle. However, the specimens with a $w_g$ of 6% and 8% exhibit exfoliation, with cracks developed after three dry-wet cycles. As $w_g$ is increased to 10%, only a bit of soil exfoliation can be observed after all six dry-wet cycles.

Above all, the change in the samples becomes relatively unapparent during the dry-wet cycle test with the increase in curing agent content, and the morphological characteristics such as corner failure, crack development, and surface shedding are constrained. As a result, it can be concluded that the water resistance of the solidified silt can be enhanced by increasing the binder content.

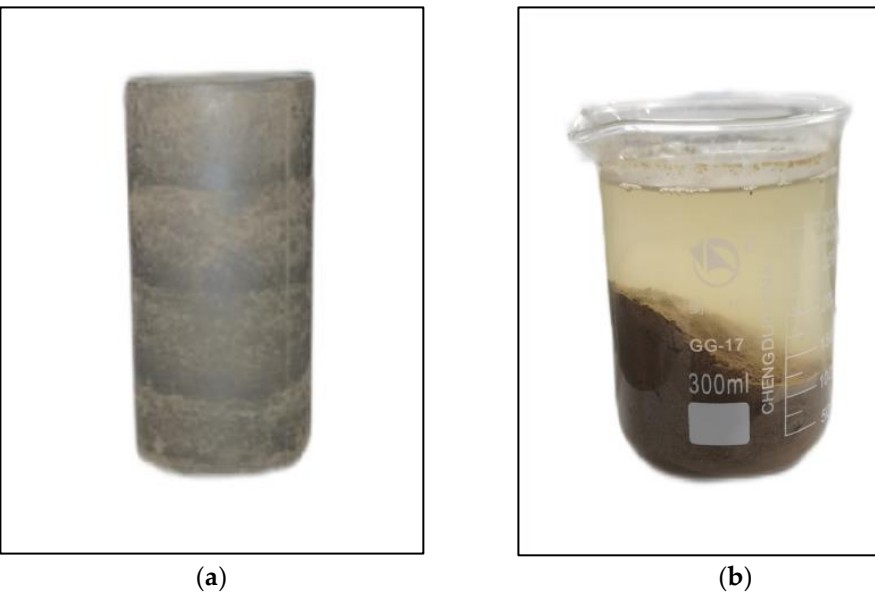

**Figure 22.** Morphology of the solidified silt with a $w_g$ of 0% during dry-wet cycles: (**a**) $N_{dw} = 0$, (**b**) $N_{dw} = 1$.

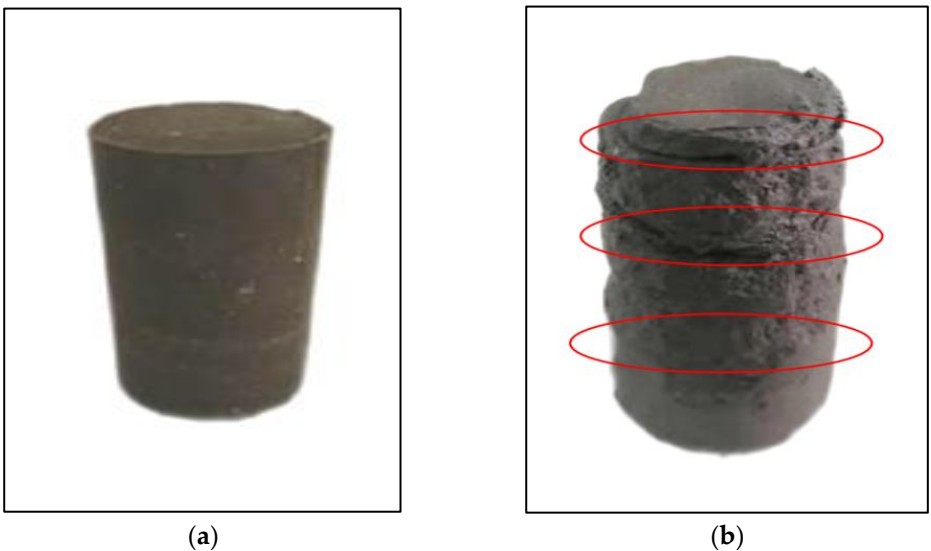

**Figure 23.** Morphology of the solidified silt with a $w_g$ of 2% during dry-wet cycles: (**a**) $N_{dw} = 0$, (**b**) $N_{dw} = 1$.

### 3.5.3. Effect of Temperature on the Water Resistance of Stabilized Silt

The above dry-wet cycle test was carried out according to the American ASTM standard, in which the temperature, $w_T$, is kept at 60 °C. However, in practical engineering, the temperature of the road subgrade filler in China is generally 30 °C more or less. Thus, to investigate the impact of temperature on the water resistance of stabilized silt, the dry-wet cycle test was first performed at 30 °C for 24 h, then cooled for 1 h and soaked in water for 23 h, and the following operation was consistent with that at 60 °C.

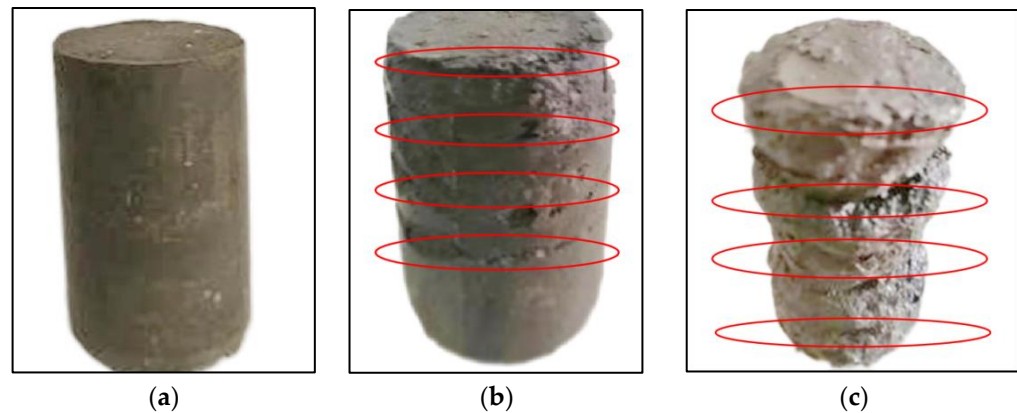

**Figure 24.** Morphology of the solidified waste silt before and after drying and wetting cycles when $w_g$ = 4%: (**a**) $N_{dw}$ = 0, (**b**) $N_{dw}$ = 1, (**c**) $N_{dw}$ = 3.

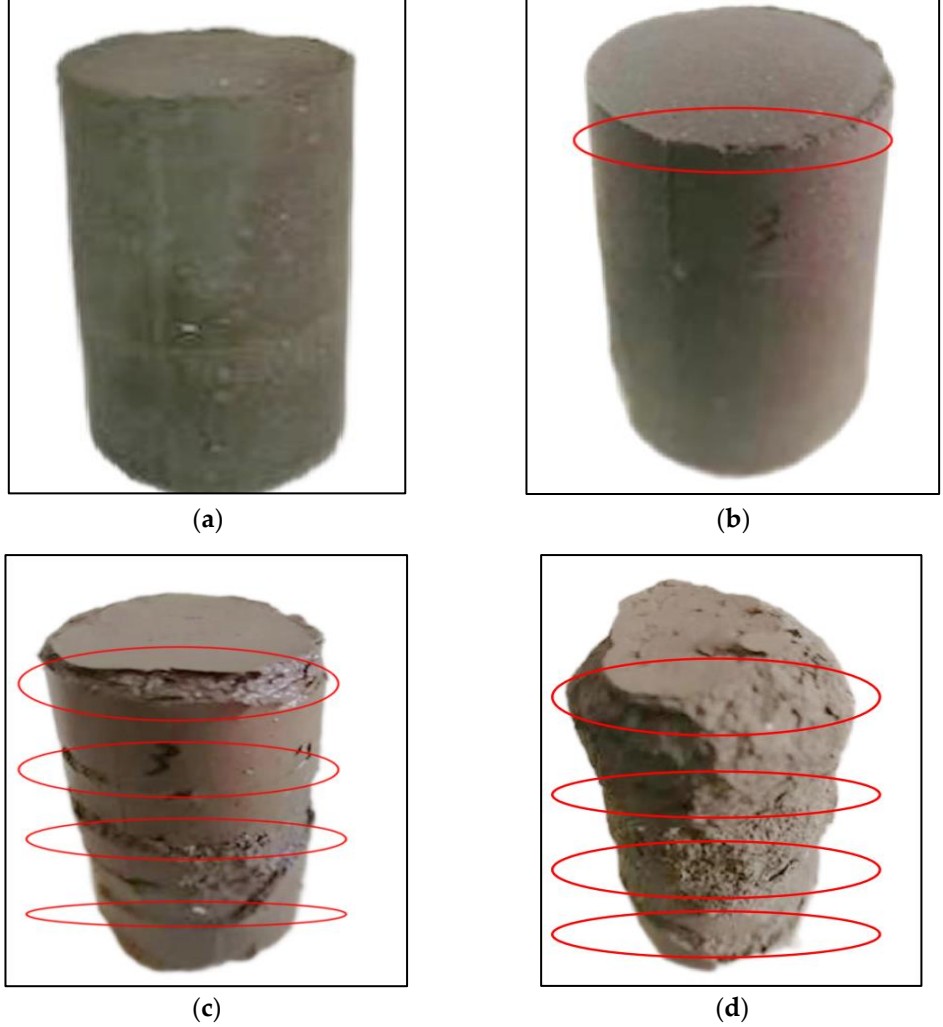

**Figure 25.** Morphology of solidified waste silt before and after drying and wetting cycles when $w_g$ = 6%: (**a**) $N_{dw}$ = 0, (**b**) $N_{dw}$ = 1, (**c**) $N_{dw}$ = 3, (**d**) $N_{dw}$ = 6.

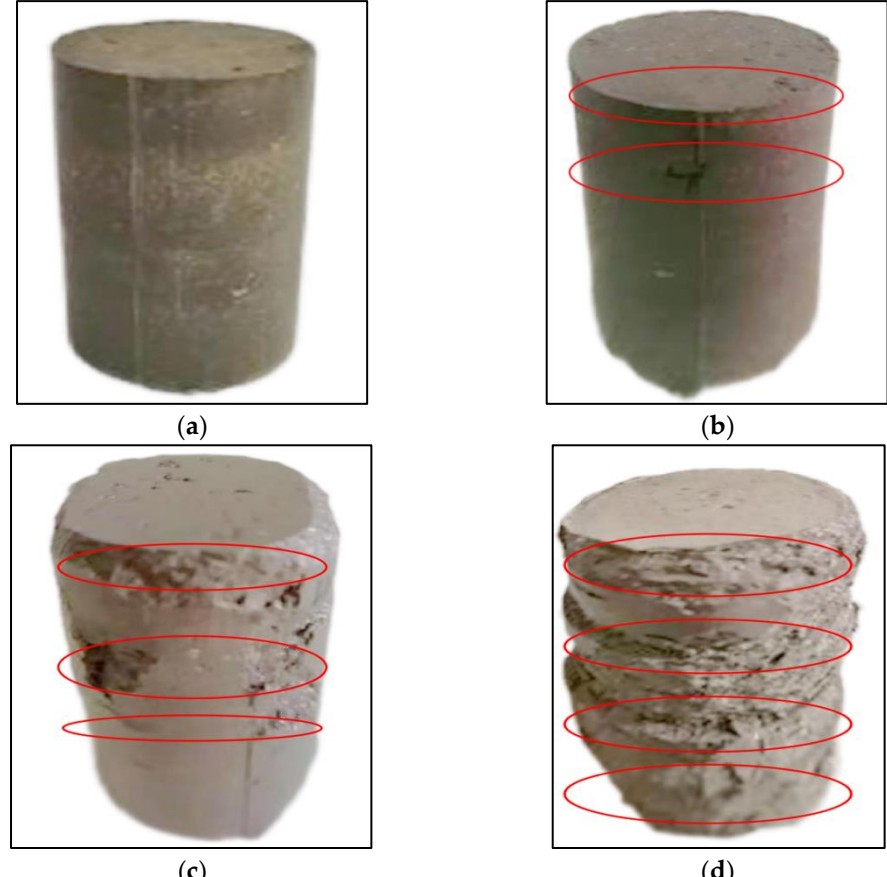

**Figure 26.** Morphology of the solidified waste silt before and after dry-wet cycles when $w_g$ = 8%: (**a**) $N_{dw}$ = 0, (**b**) $N_{dw}$ = 1, (**c**) $N_{dw}$ = 3, (**d**) $N_{dw}$ = 6.

As shown in Figure 28, in the first dry-wet cycle ($N_{dw}$ = 1), the $q_u$ of the solidified silt at 30 °C drops from 1234.642 kPa to 712.253 kPa with a 42.3% reduction, which is lower than the reduction of 52.97% at 60 °C. With the further conducting of the dry-wet cycle test, the $q_u$ of the solidified silt continues to decrease to 401.275 kPa, which is almost twice that at 60 °C. When the temperature is close to the normal temperature, 30 °C, the specimen exhibits a reduction in pores and the metamorphosis and absence of penetration and direct disintegration, which is supposed to be responsible for the upper obtained change of $q_u$. Consequently, the temperature is considered to possess a negative effect on the water resistance of solidified silt.

### 3.5.4. Effect of Organic Matter Content on the Water Resistance of Stabilized Silt

The marine engineering waste silt always contains a certain amount of organic matter, which has been considered to have a negative effect on the curing agent hydration [56]. As a result, it is necessary to study the effect of the organic matter content, $w_o$, on the water resistance of stabilized silt. As shown in Figure 29, $w_o$ presents a significant negative effect on the water resistance of stabilized soil with a $w_g$ of 6%. For each organic mass content, $q_u$ always behaves as a reduction with the increasing of the dry-wet cycles, $N_{dw}$. Moreover, the $q_u$ of the specimens with less content of organic matter is always larger than that with more content of organic matter at the same dry-wet cycles. Therefore, it can be considered that organic matter plays the role of reducing the water resistance of stabilized silt. Additionally, it is also observed that when $w_o$ is less than 4%, the solidified silt still has a specific strength after six dry-wet cycles. However, when $w_o$ exceeds 4%, the $q_u$ of the solidified silt drops significantly after six dry-wet cycles, especially when $w_o$ = 10%, at which $q_u$ drops to nearly 0.

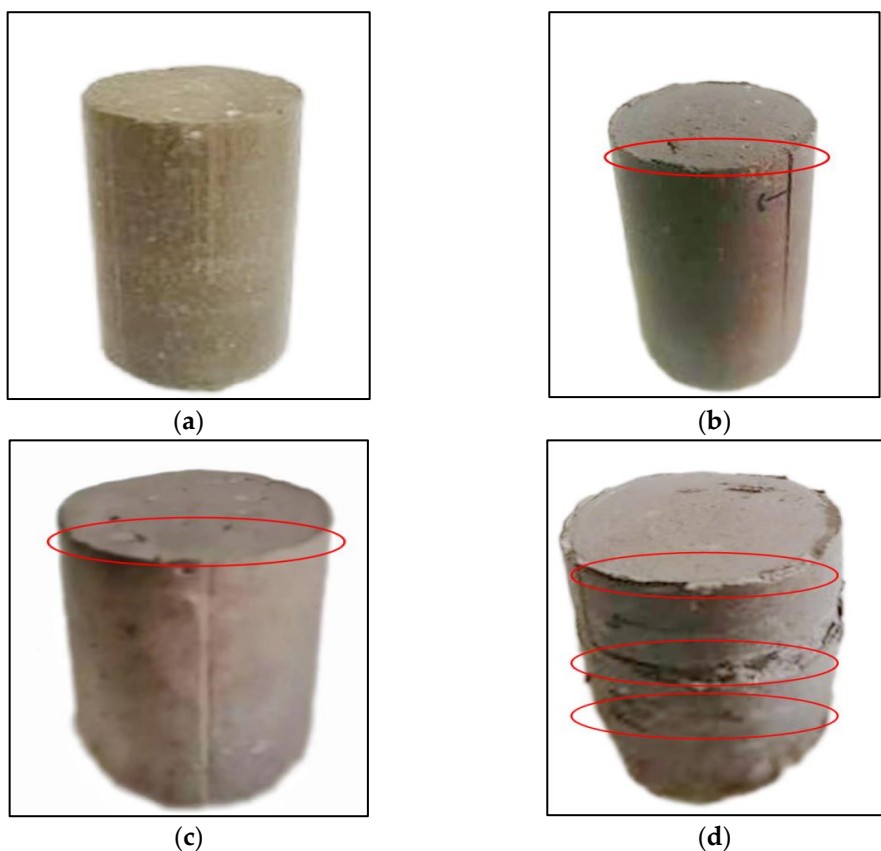

**Figure 27.** Morphology of the solidified waste silt before and after dry-wet cycles when $w_g$ = 10%: (**a**) $N_{dw}$ = 0, (**b**) $N_{dw}$ = 1, (**c**) $N_{dw}$ = 3, (**d**) $N_{dw}$ = 6.

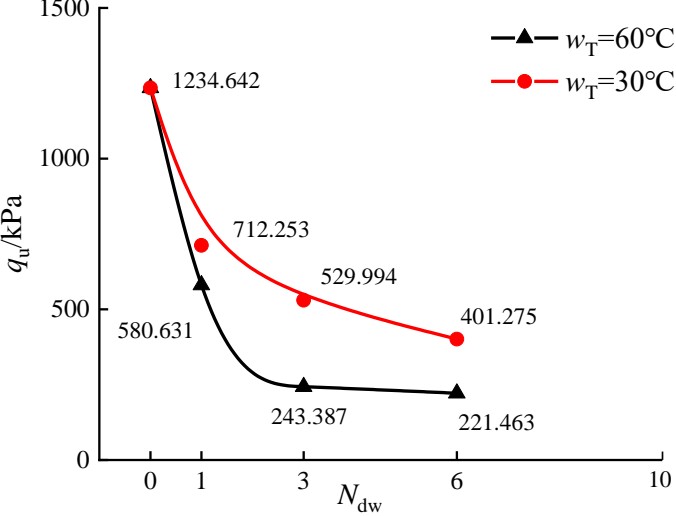

**Figure 28.** Dry-wet cycle test of the solidified waste silt with curing agent content of 6%.

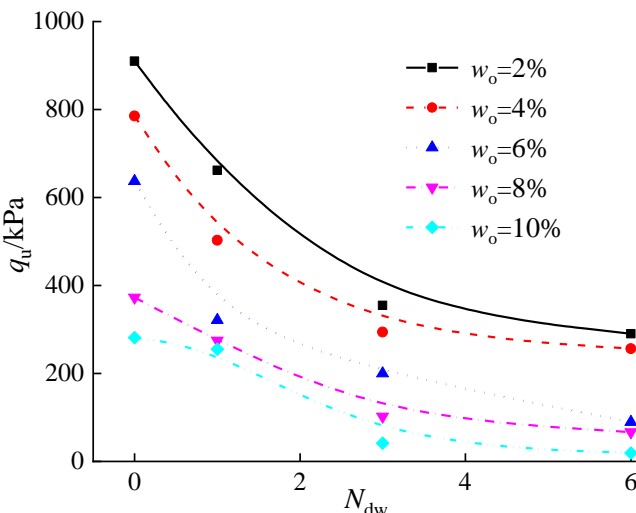

**Figure 29.** $q_u$ of stabilized silt under different dry-wet cycles and organic matter contents.

## 4. Conclusions

(1) The optimal ratio of CCR and PA addressed by PSO-SVM is CCR:PA = 13.35:16.06, and the associated compound curing agent was named PZ-1.

(2) The main stabilized mechanism of PZ-1 on marine waste silt is that the $Ca(OH)_2$ from the CCR can react with the $SiO_2$ from the PA and further active the mineral composition of waste silt to form cementitious substances such as calcium silicate hydrate (C-S-H) and Ettringite. Moreover, $Ca^{2+}$ and $Mg^{2+}$ will carbonize under an alkaline environment to form $CaCO_3$ and $CaMg(CO_3)_2$ to fill pores and cement soil particles and then enhance the soil structure.

(3) With the increase in curing agent content, the $q_u$ of the solidified silt first increases and then decreases, with the $q_u$ peak value appearing in the content range of 8–10%, which is due to the optimal reacting condition between PZ-1 and silt. However, the $q_u$ of the solidified silt begins to drop when $w_g$ exceeds 10% because of the inhibiting effect of the relatively insufficient soil water on the hydration of the redundant PZ-1.

(4) The initial damage to the specimen occurred in the corner of the specimen and the contact part of each layer in the dry-wet cycles. Generally, the stronger water resistance of the stabilized silt was obtained with the increase in the curing agent content, $w_g$, which is different from the hump-curve relationship between the $q_u$ and $w_g$ of Figure 12. It should be noted that the $q_u$ attained with a $w_g$ of 10% is larger than that with $w_g$ of 8% due to the fact that the redundant curing agent in the former case continues to hydrate and further enhance the water tolerance of the stabilized silt.

(5) Organic matter has a significant negative effect on the water resistance of stabilized soil. The water resistance of solidified soil in the actual project at about 30 °C is higher than that at 60 °C.

It should be noted that the present research mainly focuses on the effect of the dry-wet cycle on the physical and mechanical behavior of marine waste silt solidified by calcium carbide residue and plant ash, which is helpful for the application of marine engineering waste silt solidified by industrial or agricultural wastes at the early stage. In addition, the in-depth investigation on the mechanism of PZ-1 solidifying waste silt provides theoretical guidance for its application in practical engineering, which will be carried out in the future.

**Author Contributions:** Data curation, validation, writing-original draft, conceptualization, revision, H.Y.; supervision, J.Z.; project administration, Y.T.; methodology, Z.W.; resources, Q.Z. All authors have read and agreed to the published version of the manuscript.

**Funding:** This research was funded by the National Natural Science Foundation of China, grant number 51879133, and the Commonweal Project of Zhejiang Province, grant number LGG22E090002.

**Institutional Review Board Statement:** Not applicable.

**Informed Consent Statement:** Not applicable.

**Data Availability Statement:** Not applicable.

**Conflicts of Interest:** The authors declare no conflict of interest.

## Abbreviations

| | |
|---|---|
| $w_g$ | curing agent dosage |
| $w_o$ | organic matter content |
| $w$ | water content |
| $C$ | Abundance |
| $D$ | fractal dimension |
| $R$ | Roundness |
| $q_u$ | unconfined compressive strength |
| $T$ | curing time |
| $N_{dw}$ | number of dry-wet cycles |
| $w_T$ | Temperature |

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
