# Peer review of "Effect of the Dry-Wet Cycle on the Performance of Marine Waste Silt Solidified by Calcium Carbide Residue and Plant Ash"

_jmse, doi:10.3390/jmse10101442_

Round 1
Reviewer 1 Report
The article is interesting and in the focus of the journal.
The structure is clear and well presented.
Figure 8 - must be removed, no scientific value.. we all know XRD devices.
Similar to figure 9, if this kept, should be indicated the component parts.
The results are sound and the conclusions are suitable.
Author Response
Response to Reviewers’ comment (jmse-1921442)
We would like to thanks the editor, reviewers and the assessor for their considered and helpful comments on our paper. We have endeavored to answer these comments as fully as possible in our response below:
Reviewer: The article is interesting and in the focus of the journal. The structure is clear and well presented.
Comment #1: Figure 8 must be removed, no scientific value. we all know XRD devices. Similar to figure 9, if this kept, should be indicated the component parts.
Response #1: The authors agree with the reviewer’s advice. In the revised manuscript, both Figure 8 and Figure 9 had been removed.

Reviewer 2 Report
While there is interesting and original analysis and results, there are several shortcomings in the manuscript. Hence, major revision with a fresh round of review is recommended, based on the following specific comments:
1. The title should be concise. The Abstract should be concise to highlight the main subject matter and aspect of the paper with brief findings.
2. The literature review portion lacks from completeness, as numerous important and recent relevant contributions are missing, for example:
https://www.sciencedirect.com/science/article/pii/S0956053X20302944
https://doi.org/10.1155/2018/2321361
https://doi.org/10.3390/ma15144919
3. The notations are not properly explained which makes it difficult to follow the manuscript. The list of notations appended at the end of the manuscript is also incomplete.
4. The Compaction curve in Figure 12 seems to be incorrect, as the peak point may be situated higher than the correct one.
5. Page 19: The parameter R is not clearly defined.
6. Some of the experimental results could be compared with available theoretical results using existing models. This will enhance the paper quality.
7. The significance and practical application of the work requires to be highlighted in a separate heading, preferably with appropriate design recommendations.
8. The conclusion should be concise with focus on the primary research findings.
9. Minor comments:
(a) Improvement in English write-up is required.
(b) The list of notations is incomplete.
Author Response
Response to Reviewers’ comment (jmse-1921442)
We would like to thanks the editor, reviewers and the assessor for their considered and helpful comments on our paper. We have endeavored to answer these comments as fully as possible in our response below:
Reviewer: While there is interesting and original analysis and results, there are several shortcomings in the manuscript. Hence, major revision with a fresh round of review is recommended, based on the following specific comments:
Comment #1: The title should be concise. The Abstract should be concise to highlight the main subject matter and aspect of the paper with brief findings.
Response #1: Thanks for reminding, the title and the abstract have been simplified to be “Effect of the dry–wet cycle on the performance of marine waste silt solidified by calcium carbide residue and plant ash”, symbols appearing in the abstract have been explained and supplemented, moreover, the abstract had been rearranged to highlight the main subject matter and aspect of the paper with brief findings as follows [ Page 1].
…. This research aims to investigate the potential of engineering waste marine silt stabilized by a self-developed stabilizing chemical additive called PZ-1 as subgrade filler. PZ-1 is composed of calcium carbide residue (CCR) and plant ash (PA) under an optimal composition ratio determined by coupling particle swarm optimization with a support vector machine (PSO-SVM). The effect of curing agent dosage(wg), temperature(wT), number of dry-wet cycles (Ndw), and organic matter content (wo) on the micro-macro behavior of stabilized silt were investigated via the Unconfined Compressive Strength (UCS) test, Scanning Electron Microscope (SEM) test, and X-ray diffraction (XRD) test. The experimental results demonstrate a significant positive effect of PZ-1 on the unconfined compressive strength (qu) of marine engineering waste silt with curing agent contents of 0% ~ 8%. It is also found that strength improvement of the stabilized silt can be attributed to the formation of gelling substances such as C-S-H and calcite. Water resistance of the stabilized silt can be enhanced by increasing the dosage of the curing agent. Moreover, organic matter content and ambient temperature have significant effects on the dry-wet cycle tolerance of solidified soil, among which temperature exhibits more obvious.….
Comment #2: The literature review portion lacks from completeness, as numerous important and recent relevant contributions are missing, for example:
https://www.sciencedirect.com/science/article/pii/S0956053X20302944
https://doi.org/10.1155/2018/2321361
https://doi.org/10.3390/ma15144919
Response #2: The authors agree with the reviewer’s opinion. The literatures of Çevikbilen et al. (2020), Ye et al. (2018) and Govedarica et al. (2022) had been cited in the revised manuscript in blue color as follows:
…. At present, the most popular stabilized agent in soil improvement are lime, Portland cement, and their compound types [4–7]….
…. In this respect, using various types of solid wastes as industrial waste and agricultural waste to substitute for high-carbon-emission stabilized agents, such as lime, Portland cement or their compound types, was considered a fairly effective strategy [18-19].….
The associated reference had been added in the reference list in the revised manuscript in blue color as follows[ Pages 26- 27]:
….[7] Çevikbilen, G.; BaÅŸar, H. M.; KaradoÄŸan, Ü.; Teymur, B.; DaÄŸlı, S.; Tolun, L. Assessment of the use of dredged marine materials in sanitary landfills: A case study from the Marmara sea. Waste Management. 2020, 113, 70-79.….
….[18] Ye, H.; Chu, C. F.; Xu, L.; Guo, K. L.; Li, D. Experimental studies on drying-wetting cycle characteristics of expansive soils improved by industrial wastes. Advances in Civil Engineering. 2018, 2321361(9).
….[19] Govedarica, O.; Aškrabić, M.; HadnaÄ‘ev-Kostić, M.; Vulić, T.; Lekić, B.; Rajaković-Ognjanović, V.; Zakić, D. Evaluation of solidified wastewater treatment sludge as a potential SCM in pervious concrete pavements. Materials. 2022, 15, 4919.….
….[50] Chindaprasirt, P.; Kampala, A.; Jitsangiam, P.; Horpibulsuk, S. Performance and evaluation of calcium carbide residuest abilized lateritic soil for construction materials. Case Studies in Construction Materials. 2020,13(10) .….
Comment #3: The notations are not properly explained which makes it difficult to follow the manuscript. The list of notations appended at the end of the manuscript is also incomplete.
Response #3: Thanks for reminding, the list of notations had been appended at the end of the manuscript as follows[ Page 26]:
….
Notations:
wg curing agent dosage
wo organic matter content
w water content
C abundance
D fractal dimension
R roundness
qu unconfined compressive strength
T curing time
Ndw number of dry–wet cycles
wT temperature
….
Comment #4: The Compaction curve in Figure 12 seems to be incorrect, as the peak point may be situated higher than the correct one.
Response 4: Thanks for reminding, the Compaction curve had been refitted in the revised manuscript as follows [Page 13]:
….
Fig. 10 Compaction curve of natural waste silt
….
Comment #5: Page 19: The parameter R is not clearly defined.
Response #5: The parameter R presents the roundness of soil particle, which had been defined in section 2.3.9[Page 11].
Comment #6: Some of the experimental results could be compared with available theoretical results using existing models. This will enhance the paper quality.
Response #6: The authors agree with the reviewer’s opinion. The relationship between qu and curing agent content has been compared with the linear function founded by Prinya et al. (2020). Moreover, the relationship between qu and the number of the dry-wet cycle has been compared with the prediction model of Ye et al. (2018). The discovery in the present experimental results had been enhanced in the revised manuscript as follows [Page 14 and Page 19]:
…. As shown in Fig. 12, curing agent dosage, wg, plays a significant role in the strength development of solidified silt. With the increase of wg from 0 to 8%, qu increases from 410.354 kPa to 1403.447 kPa with a sharp growth rate of 242.01%, and the relationship between qu and the content of the curing agent can be expressed by a linear function when the hydration reaction can be fully carried out, which is consistent with the model developed by Prinya et al [50].….
…. Moreover, qu of the specimen with a curing time of 28 days conforms to the exponential relationship with the evolution of the dry-wet cycle, which is consistent with the prediction model developed by Ye et al [18].….
Comment #7: The significance and practical application of the work requires to be highlighted in a separate heading, preferably with appropriate design recommendations.
Response #7: This is a good suggestion. The present research is a fundamental work on the improving mechanism of waste marine silt solidified with PZ-1 and the effect of dry-wet cycle on the physical and mechanical behavior of PZ-1 stabilized silt, which provides theoretical guidance for its future application in practical engineering. The following significance of the work and future plan had been added in the revised manuscript as follows[ Page 26]:
…. It should be noted that the present research mainly focuses on the effect of the dry-wet cycle on the physical and mechanical behavior of marine waste silt solidified by calcium carbide residue and plant ash, which is helpful for the application of marine engineering waste silt solidified by industrial or agricultural wastes at the early stage. In addition, the in-depth investigation on the mechanism of PZ-1 solidifying waste silt provides theoretical guidance for its application in practical engineering, which will be carried out in the future.….
Comment #8: The conclusion should be concise with focus on the primary research findings.
Response #8: As suggested by the reviewer, the conclusion has been simplified in the revised version as follows [ Page 26]:
…. (2) The main stabilized mechanism of PZ-1 on marine waste silt is that the Ca(OH)2 from CCR can react with SiO2 from PA and further active mineral composition of waste silt to form the cementitious substance such as calcium silicate hydrate (C-S-H) and Ettringite. Moreover, Ca2+ and Mg2+ will carbonize under an alkaline environment to form CaCO3 and CaMg(CO3)2 to fill pores and cement soil particles and then enhance the soil structure.….
…. (3) With the increase of curing agent content, qu of the solidified silt first increases and then decreases with the qu peak value appearing in the content range of 8% –10%, which is due to the optimal reacting condition between PZ–1 and silt. However, qu of the solidified silt begins to drop when wg exceeds 10% because of inhibiting effect of the relative insufficient soil water on the hydration of redundant PZ–1.….
…. (4) The initial damage to the specimen occurred in the corner of the specimen and the contact part of each layer in dry-wet cycles. Generally, stronger of the water–resistance of stabilized silt was obtained with the increase of curing agent content, wg, which is different from the hump–curve relationship between qu and wg of Fig.12. It should be noted that qu attained with wg of 10 % is larger than that with wg of 8 % due to that the redundant curing agent in the former case continues to hydrate and further enhance the water tolerance of stabilized silt.….
…. (5) Organic matter has a significant negative effect on the water resistance of stabilized soil. The water resistance of solidified soil in the actual project at about 30 °C is higher than that at 60 °C.….
Comment #9: Minor comments:
(a) Improvement in English write-up is required.
(b) The list of notations is incomplete.
Response #9: As suggested by the reviewer, English statements have been improved in blue mark and the gloss list has been refined.

Round 2
Reviewer 2 Report
May be accepted.